# Sensitivity of the surface energy budget to drifting snow as simulated by MAR in coastal Adelie Land, Antarctica

Louis Le Toumelin[1, 2], Charles Amory[1,3], Vincent Favier[1], Christoph Kittel[3], Stefan Hofer[4],
Xavier Fettweis[3], Hubert Gallée[1], and Vinay Kayetha[5]

[1]Université Grenoble Alpes, CNRS, Institut des Géosciences de l'Environnement, 38000, Grenoble, France
[2]Univ. Grenoble Alpes, Université de Toulouse, Météo-France, CNRS, CNRM, Centre d'Études de la Neige, Grenoble, France
[3]F.R.S.-FNRS, Laboratory of Climatology, Department of Geography, University of Liège, 4000 Liège, Belgium
[4]Department of Geosciences, University of Oslo, Oslo, Norway
[5]Science Systems and Applications, Greenbelt, MD, USA

**Correspondence:** Louis Le Toumelin (louis.letoumelin@gmail.com)

**Abstract.** In order to understand the evolution of the climate of Antarctica, dominant processes that control surface and low-atmosphere meteorology need to be accurately captured in climate models. We used the regional climate model MAR (v3.11) at 10 km horizontal resolution, forced by ERA5 reanalysis over a 9-year period (2010–2018), to study the impact of drifting snow (designing here the wind-driven transport of snow particles below and above 2 m) on the near-surface atmosphere and surface in Adelie Land, East Antarctica. Two model runs were performed, respectively with and without drifting snow, and compared to half-hourly in situ observations at D17, a coastal and windy location of Adelie Land. We show that sublimation of drifting-snow particles in the atmosphere drives the difference between model runs and is responsible for significant impacts on the near-surface atmosphere. By cooling the low atmosphere and increasing its relative humidity, drifting snow also reduces sensible and latent heat exchanges at the surface (- 5.7 $Wm^{-2}$ on average). Moreover, large and dense drifting-snow layers act as near-surface cloud by interacting with incoming radiative fluxes, enhancing incoming longwave radiations and reducing incoming shortwave radiations in summer (net radiative forcing: 5.7 $Wm^{-2}$). Even if drifting snow modifies these processes involved in surface-atmosphere interactions, the total surface energy budget is only slightly modified by introducing drifting snow, because of compensating effects in surface energy fluxes. The drifting-snow driven effects are not prominent near the surface but peak higher in the boundary layer (fourth vertical level, 12m) where drifting snow sublimation is the most pronounced. Accounting for drifting snow in MAR generally improves the comparison at D17, more especially for the representation of relative humidity (mean bias reduced from -14.0 % to -0.7 %) and incoming longwave radiation (mean bias reduced from -20.4 $Wm^{-2}$ to -14.9 $Wm^{-2}$). Consequently, our results suggest that a detailed representation of drifting-snow processes is required in climate models to better capture the near–surface meteorology and surface—atmosphere interactions in coastal Adelie Land.

## 1 Introduction

In order to improve estimates of the contribution of the Antarctic ice sheet to sea level rise in a global warming scenario (Edwards et al., 2019; Shepherd et al., 2018), an accurate representation of the current surface mass balance (SMB) of the ice

sheet and overlying atmospheric physics in models is necessary (Agosta et al., 2019; van Wessem et al., 2018). A particular feature of the climate of Antarctica is the widespread, wind-driven removal and transport of snow, often referred to as drifting and blowing snow. Both processes are theoretically distinguished by the height of the wind-driven snow particles (below 2 m for drifting snow and above that height for blowing snow). For convenience, in our study drifting and blowing snow are combined into the single term of drifting snow.

Locally, drifting snow has proven itself a key SMB parameter. Even if drifting snow is subject to a high spatial and temporal variability, significant yearly frequency (up to > 90 % of the time) and mass transport values have been reported at various places scattered over the Antarctic continent (e.g., Gossart et al., 2017; Mahesh et al., 2003; Mann et al., 2000; Scarchilli et al., 2010; Amory, 2020), especially in the megadune region and coastal windy regions (Palm et al., 2017, 2018). Over coastal locations, wind-driven ablation (erosion and sublimation of drifting-snow particles) and precipitation can be of the same order of magnitude (Scarchilli et al., 2010; van den Broeke et al., 2006). Drifting snow can spread horizontally over hundreds of kilometers, vertically over hundreds of meters (Palm et al., 2011) and remove by erosion all the accumulated firn at the surface, creating climate-sensitive low-albedo blue-ice areas (Bintanja, 1999; Favier et al., 2011; Scarchilli et al., 2010). At the scale of the Antarctic ice sheet, model studies even suggest that ablation may be primarily due to drifting snow (Lenaerts and van den Broeke, 2012; van Wessem et al., 2018; Palm et al., 2018), although drifting-snow mass transport could still be underestimated in regional-model-based estimates of the Antarctic SMB (Agosta et al., 2019). Despite these efforts, drifting-snow processes still need to be more accurately resolved in models and better observationally constrained to improve our understanding of their influence on the climate and surface mass balance of Antarctica (Favier et al., 2017; Amory, 2020; Hanna et al., 2020).

Drifting-snow particles influence the local climate through their interactions with the lower atmosphere and the surface energy budget (SEB). Latent heat consumption and moisture release through sublimation of wind-driven particles modify the vertical gradients in temperature and humidity (e.g., Schmidt, 1982; Déry et al., 1998; Bintanja, 2000; Amory and Kittel, 2019), further affecting the turbulent heat exchange with the surface (Bintanja, 2001; Lenaerts and van den Broeke, 2012; Barral et al., 2014). Yang et al. (2014) observed through remotely sensed data that drifting snow can increase top-of-atmosphere outgoing longwave radiation by more than 20 $Wm^{-2}$ during wintertime in East Antarctica, suggesting a significant contribution of drifting snow to the atmospheric radiative budget. In the cold environment of central Antarctica, the lower atmosphere is usually very dry and clouds are generally optically thin (Mahesh et al., 2003; Town et al., 2007). The SEB is thus particularly sensitive to increases in the atmospheric longwave emissivity caused by additional suspended particles or water vapour due to drifting-snow sublimation. Yamanouchi and Kawaguchi (1984) highlighted increases in downwelling longwave radiation up to 20 $Wm^{-2}$ below 30 m above ground during drifting snow from observations collected at Mizuho Station. The occurrence of drifting-snow layers has been linked to increases in surface temperature of typically a few degrees at South Pole (Mahesh et al., 2003). In a modelling study with the regional climate model MAR, Gallée and Gorodetskaya (2010) showed that neglecting the contribution of suspended snow particles to the atmospheric longwave emissivity resulted in underestimation of the surface temperature at Dome C. Only small rises in surface temperature can be expected on the Antarctic Plateau in response to additional drift-induced radiative forcing due to the strong surface-based temperature inversion that prevails throughout the year. But over coastal areas of the Antarctic ice sheet, which experience stronger wind speeds and related turbulent mixing,

and where higher, optically thicker drifting-snow layers can frequently develop (Palm et al., 2018), radiative effects of drifting snow remain currently poorly documented.

Drifting-snow data are extremely limited over high-latitude regions and still remain challenging to collect in the extreme and remote Antarctic environment (Amory, 2020). Drifting-snow effects are moreover directly embedded in measurable climate quantities and can hardly be disentangled from usual atmospheric measurements without an accurate knowledge of drifting-snow properties in the whole atmospheric column. As an alternative, regional climate models provide continuous, high-resolution gridded estimates of individual climate components over large areas (van Wessem et al., 2018; Agosta et al., 2019). Detailed modelling may thus provide physical insights into the relevance and climatological significance of drifting snow. However, only a few regional models currently explicitly quantify drifting-snow processes (Lenaerts and van den Broeke, 2012; Gallée et al., 2013; Amory et al., 2015, 2021), and different implementation strategies from one model to another have been employed to account for interactions of drifting snow with the atmosphere (Gallée et al., 2013).

In this paper we use the regional climate model MAR to quantify the influence of drifting snow on the near-surface climate and SEB in Adelie Land, a coastal region of East Antarctica particularly prone to erosive winds and where drifting-snow equipment deployed over the past few years provide observational support for model evaluation near the surface (Trouvilliez et al., 2014; Amory et al., 2021). MAR has been widely used to simulate the climate and surface mass balance of polar ice sheets (e.g., Fettweis et al., 2017, 2020; Hofer et al., 2017, 2019; Kittel et al., 2018, 2021; Mottram et al., 2020), and includes a detailed representation of drifting-snow processes already applied to study snow mass transport and wind-driven ablation in coastal East Antarctica (Gallée et al., 2005, 2013; Amory et al., 2015, 2021). The explicit coupling of the drifting-snow scheme with the atmospheric component of the model enables a vertical discretization of drifting-snow profiles and related sublimation within the atmospheric boundary layer and takes into account the radiative contribution of drifting-snow particles.

Observations, model setup and data processing methods are described in Sect. 2. The main modifications induced by drifting snow on the surface and near-surface meteorology at D17 are detailed in Sect. 3. Sect. 4 discusses the results including the impact of drifting snow on the boundary layer and their spatial distribution in Adelie Land. Finally, Sect. 5 summarizes and concludes the study.

## 2   Methods and data

### 2.1   Field area and instrumentation

Site D17 (66.7° S, 139.9° E; 450 m above sea level, Fig. 1) is located 10 km inland and 15 km southwest of the permanent French station of Dumont d'Urville, in Adelie Land, East Antarctica. The measurement area is characterized by strong and persistent katabatic winds mostly originating from the south-east direction and flowing over a permanent snow surface, favouring the regular occurrence of drifting snow (Amory, 2020).

A 7 m meteorological mast has been installed at D17 in 2010 providing relative humidity, wind speed, and temperature measurements at six logarithmically spaced levels (nominal heights 0.8, 1.3, 2, 2.8, 3.9, and 5.5 m above the surface). An ultrasonic depth gauge measures changes in elevation above ground level (a.g.l.) since December 2012. Relative humidity

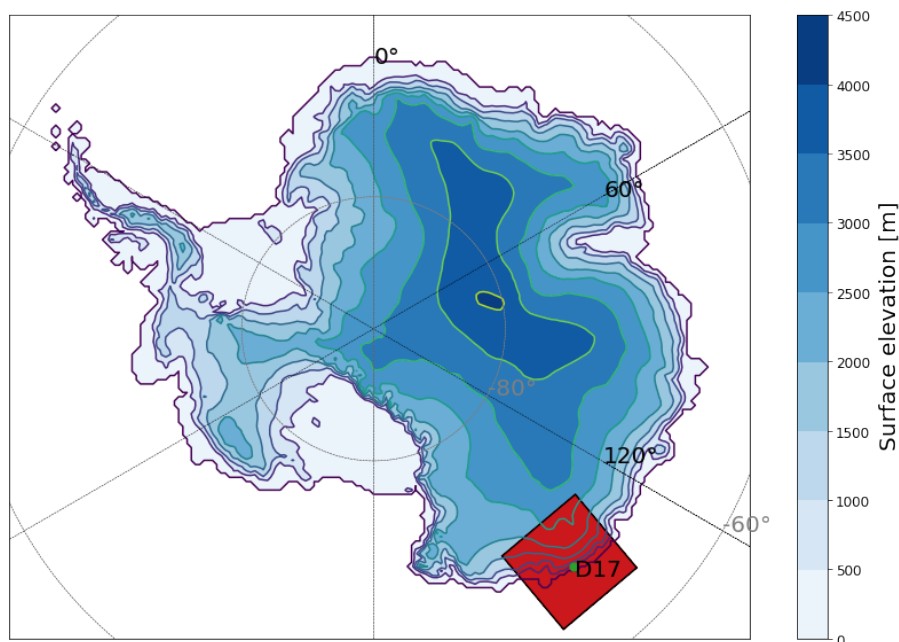

**Figure 1.** Antarctic topography as used in MAR. The integration domain over Adelie Land is displayed in red with a green dot for D17 location.

is initially assessed with respect to liquid water, and calculation necessary to convert raw values into relative humidity with respect to ice is performed according to Goff and Gratch (1945). As supersaturation is very unlikely at this coastal location notably due to frequent drifting snow, converted values exceeding 100 % are attributed to limitation of the conversion method and capped to 100 % (Barral et al., 2014).

Radiative fluxes designate the incoming shortwave radiation (SWD), outgoing shortwave radiation (SWU), incoming longwave radiation (LWD) and outgoing longwave radiation (LWU). All fluxes are defined positive when directed towards the surface. A Kipp and Zonen CNR4 net radiometer has been installed in early February 2014 next to the meteorological mast (Amory et al., 2020b). This sensor is composed by two pairs of pyrgeometers and pyranometers: the first one measures SWD and SWU (spectral range: 300 to 2800 nm) and the second one LWD and LWU (spectral range 4500 to 42000 nm). Sensor characteristics are presented in Table 1. Negative values of each radiation flux were set to 0. Summer maintenance operations and winter excessive discharge of the station's battery between May and September 2018 impacted measurement continuity. The later period was characterized by a gap of 55 % in the radiative flux observations. Outside this last time frame, few observations are missing as reported in Table 1.

Calculation of the turbulent heat fluxes, composed by the sensible heat flux (SHF) and the latent heat flux (LHF), could be possible with observations at D17 (relative humidity, temperature and wind speed) through the application of profile method (Barral et al., 2014). However, a first concern can be raised about the applicability of Monin–Obukhov similarity theory in

drifting-snow conditions, as the requirement of vertical constancy in turbulent fluxes is not met (Bintanja, 2001; Amory and Kittel, 2019). Moreover, during drifting-snow occurrences, turbulent mixing and atmospheric sublimation favour the establishment of near-surface atmospheric layers characterized by low vertical gradients in humidity and temperature. Those gradients are frequently lower than the sensor accuracies, leading to large uncertainties in the derived turbulent fluxes. Barral et al. (2014) observed amplified uncertainty with strong winds at D17, furthermore limiting the use of the profile method during drifting-snow conditions. The same conclusion is drawn here, as determination of turbulent fluxes has been intended but led to strongly diverging results according to the choice of the method (bulk vs profile method), the stability function, the number and the considered height a.g.l. of the measurement levels (not shown). Thus, no observed turbulent fluxes are available here for model evaluation.

Information of drifting snow is obtained using second-generation IAV engineering FlowCapt sensors (hereinafter referred to as 2G-FlowCapt™, Amory et al. (2020a)). The 2G-FlowCapt™ is a 1 m long tube, containing an electroacoustic transducer measuring the noise generated by the impact of drifting-snow particles on the tube. The signal is then converted into a drifting-snow flux integrated over the exposed length of the tube. At D17, a pair of 2G-FlowCapt™ instruments has been operational since late December 2012. The sensors are set up vertically one above the other starting from the ground. This configuration enables the detection of the initiation of drifting snow and measurement of the drifting-snow flux near the surface ($\leq 2$ m).

While the 2G-FlowCapt™ has been shown to underestimate drifting-snow fluxes compared to optical measurements in the French Alps (Trouvilliez et al., 2015), its behavior still needs to be assessed in the extreme Antarctic environment where different climatic conditions and particle properties influencing the measurement can be expected (Cierco et al., 2007). A preliminary evaluation of the 2G-FlowCapt™ instrument against optical measurements has been performed at D17 during one drifting-snow event and shows good agreement between both types of sensors (Amory, 2020). The lowest 2G-FlowCapt™ (respectively the highest), of exposed length $h_1$ (respectively $h_2$), measures a drifting-snow flux designated as $FC_1$ (respectively $FC_2$). Eq. (1) expresses the mean drifting-snow flux FC and takes into account the measurement height as accumulation can partially bury the lower 2G-FlowCapt™:

$$FC = \frac{h_1 \cdot FC_1 + h_2 \cdot FC_2}{h_1 + h_2} \tag{1}$$

All measurements are recorded every 15 s and mean values are performed every 30 min and stored on a Campbell CR3000 data logger.

## 2.2 Model description

MAR is a hydrostatic regional climate model solving primitive equations as originally described in Gallée and Schayes (1994), and has been extensively used for decade-long climate simulation over high-latitude regions (e.g., Agosta et al., 2019; Fettweis et al., 2017, 2020; Mottram et al., 2020; Kittel et al., 2021). Five atmospheric water species are represented in the model: specific humidity, cloud droplets, rain drops, cloud ice crystals, and snow particles (Gallée and Schayes, 1994). Radiative transfer through the atmosphere is calculated according to Morcrette (2002), and cloud radiative properties are calculated

**Table 1.** Observed variables and technical specifications for sensors used at D17.

| Variable | Sensor | Manufacturer | Accuracy | Observation period | $\frac{\text{Number of unaivailable data}}{\text{All time steps}}$ |
|---|---|---|---|---|---|
| Wind speed | A100LK | Campbell scientific | $1\% \pm 0.1 m s{-1}$ | 2010-2018 | <1 % |
| Relative Humidity | HMP45A | Vaisala | 3 % for RH > 90 % | 2010–2018 | <1 % |
| | | | 2 % for RH < 90 % | | |
| Temperature | HMP45A | Vaisala | $\pm 0.4°$ C | 2010-2018 | <1 % |
| Snow height | Acoustic depth gauge SR50 | Campbell scientific | $\pm 0.01$ m | 2013-2018 | <1 % |
| Drifting-snow flux | 2G-FlowCapt™ | IAV Engineering | Not specified | 2013–2018 | <1 % |
| Radiation | CNR4 | Kipp and Zonen | 5 % in daily totals | Feb. 2014–2018 | 6 % |

according to Ebert and Curry (1992) based on water species concentrations. MAR is coupled to the surface scheme SISVAT (Soil Ice Snow Vegetation Atmosphere Transfer; Gallée and Duynkerke (1997); De Ridder and Gallée (1998); Gallée et al. (2001)), which handles energy and mass transfer between the atmosphere and the surface, and includes a multi-layer snow/ice model representing snow properties (dendricity, sphericity and size) taken from an early version of the CROCUS snow model (Brun et al., 1992). Surface sublimation (which is distinguished in the model from atmospheric sublimation) and latent heat exchanges at the surface are computed following a bulk flux formulation in SISVAT.

MAR includes a drifting-snow scheme originally described in Gallée et al. (2001). A detailed description of its latest version (including updates, changes relative to the original version and interactions with the surface and the atmosphere) can be found in Amory et al. (2021). In brief, the drifting-snow scheme simulates erosion at every grid cell in which the modelled friction velocity, $u_*$, exceeds a threshold value, $u_{*t}$, depending on the local surface snow density. While former parameterisations of $u_{*t}$ in the model did involve other snow microstructural properties such as snow grain shape and size (Gallée et al., 2001) for which observations are virtually non-existent in Antarctica, here the formulation for $u_{*t}$ has been simplified and sensitivity parameters have been reduced to surface snow density only, a variable better observationally constrained (Amory et al., 2021). Once removed from the snowpack, eroded snow is mixed with the pre-existing windborne snow mass and advected to higher atmospheric levels and/or downwind grid cells by the turbulence and microphysical schemes. Interactions with the atmosphere are computed by the microphysical and the radiative transfer schemes. More particularly, atmospheric sublimation (including both cloud-originating particles and drifting-snow particles) is computed by the model microphysics (Gallée, 1995). This formulation is based on the assumption of an exponential distribution for particle size $n_s$ (Eq. 2):

$$n_s = n_0 exp(-\lambda_s D_s) \tag{2}$$

$n_0$ being a constant representing the intercept parameter of the distribution. $n_0$ is empirically determined and was set to $3*10^8 \, m^{-4}$ in our study. $D_s$ corresponds to the particle diameter (expressed in $m$) and $\lambda_s$ the dispersion parameter (expressed in $m^{-1}$. $\lambda_s$ is determined as followed (Eq. 3):

$$\lambda_s = (\frac{\pi \rho n_0}{\rho_a q_s})^{1/4} \tag{3}$$

with $\rho$ the snow particle density ($100 \, kg \, m^{-3}$), $\rho_a$ is the air density ($kg \, m^{-3}$) and $q_s$ the snow particle ratio (expressed in kg of snow per kg of air).Sublimation is then computed as a function of the air temperature, snow particle ratio and relative humidity, so that sublimation only occurs in a subsaturated environment, with respect to ice (Lin et al. (1983), their Eq. 31, p. 1072). It also considers snow particles as graupel-like snow of hexagonal type (Locatelli and Hobbs, 1974). Consequently, drifting-snow sublimation modifies the local humidity budget, the lower atmosphere stratification and moist air advection. Representing the contribution of drifting-snow layers to the atmospheric radiative forcing is accounted for in MAR by including suspended snow particles in the computation of cloud radiative properties (Gallée and Gorodetskaya, 2010). Ultimately, the momentum balance of the boundary layer is mainly affected through three distinct processes when accounting for drifting snow in MAR. Firstly, the increase in air density due to the weight of suspended snow, which is accounted for in the model by including the contribution of suspended snow in the computation of the potential virtual temperature (Gallée et al., 2001), is inherently amplified when eroded particles contribute to the airborne snow mass. Secondly, drifting-snow sublimation and subsequent cooling of the atmosphere is computed at each model vertical level and contributes to increase air density and atmospheric stability, which enhances the along-slope pressure gradient force and is a positive feedback in katabatic flows (Kodama et al., 1985; Gallée, 1998). Finally, the aerodynamic roughness length $z_0$ is computed following a relationship previously fitted on observed $z_0$ values in order to take into account the seasonality of surface roughness in a drifting-snow climate as observed in Adelie Land (Amory et al., 2021). More precisely, $z_0$ is computed as a function of air temperature (for temperature above -20°C) and set to a constant value of $2*10^{-4}$ m representative of inland conditions (Vignon et al., 2017) for temperatures below -20°C.

We used the latest model version MARv3.11 (hereinafter referred to as MAR) and setup as presented in Amory et al. (2021), in which the model is run with a horizontal resolution of 10 km over a domain of 80 x 80 grid cells centered on D17 location. The atmosphere is described with 24 levels in the vertical, with a higher vertical resolution in the low troposphere. The lowest level is situated at 2 m a.g.l. Top-of-atmosphere and lateral forcing plus sea surface conditions are taken from 6-hourly ERA5-reanalysis (Hersbach et al., 2020). ERA5 products, evaluated in Antarctica (e.g.Gossart et al. (2017)), notably assimilate radiosoundings operated every day at the closeby permanent station Dumont D'Urville, favouring a consistency between ERA5 and the observed climate in Adelie Land. Two models runs were performed with MAR over 2010–2018. In the first run (referred to as MAR-DR), the drifting-snow scheme was activated oppositely to the second run (referred to as MAR-nDR).

## 2.3 Using CALIPSO to calculate drifting-snow height in MAR

Estimates of drifting-snow layer heights in MAR-DR are calibrated on CALIPSO observations. We underline the fact that satellite products are not used here for model evaluation, but rather as an independent product from which an objective criterion can be used to infer drifting-snow layer heights in our MAR simulations.

Palm et al. (2011) developed a remotely-sensed technique to detect drifting-snow properties and particularly the drifting-snow layer height. Lidar backscattered signal interaction with drifting snow is studied using the Cloud-Aerosol Lidar and Infrared Pathfinder Satellite Observations (CALIPSO) satellite. Under clear-sky condition, an algorithm, extensively detailed in Palm et al. (2011), analyzes the CALIPSO lidar attenuated backscatter signal over Antarctica and determines elevations of a scattering layer representative of the top of a drifting-snow layer. Such estimates enable drifting-snow detection for layers higher than 30 m. However, the snow particle ratio (which equals the mass of snow particles per kg of air, including dry air, humidity and the mass of all other hydrometeors) at the top of the drifting-snow layer, is not known.

The calibration algorithm works as follows: we firstly studied CALIPSO swaths above a 1 by 1 degree box centered on D17. When the satellite swath is covering this box and both MAR and the CALIPSO detection algorithm indicate a drifting-snow occurrence, the remotely-sensed drifting-snow layer height is retrieved. Then, the snow particle ratio (the mass of snow particles per kg of air at each model vertical level) from the closest vertical level in the MAR-DR simulation is stored (referred to as $q_{s0}$). Between January 2010 and October 2017, CALIPSO detected 56 distinct drifting-snow occurrences among the 462 observations available in the D17 area, giving $q_{s0}$ values among which a mean snow particle ratio $\bar{q_{s0}}$ is computed. $\bar{q_{s0}}$, referred to as CALIPSO snow particle ratio threshold, is representative of the snow concentration required for the satellite to detect a drifting-snow layer. Secondly, all MAR-DR simulations are selected when drifting snow is simulated (i.e., when the drifting-snow flux at the lowest model level $> 10^{-3} \, kg \, m^2 \, s^{-1}$, calculated accordingly to Amory et al. (2015)). The highest vertical level with a snow particle ratio above $\bar{q_{s0}}$ corresponds to the drifting-snow layer height. In order to avoid accounting for modelled advected precipitation or atmospheric clouds as drifting-snow layers, data were filtered, according to the method described in Sect. 2.4. The model vertical discretization sets limits to the estimation of drifting-snow layer heights that are necessarily underestimated in MAR-DR if we consider CALIPSO-detected heights as a reference (drifting-snow height distributions are proposed in Supplementary material, Fig. S5).

## 2.4 Data Filtering

When drifting snow occurs, MAR computes radiative modifications related to both the presence of drifting-snow particles and the changes in the cloud representation. In Sect. 3.3, we focus on the radiative contribution of snow particles resulting from the erosion of the surface only, a task that required to filter the data.

Firstly, because cloud formation might be influenced by drifting snow through atmospheric sublimation and changes in the amount and distribution of atmospheric water species, it may induce radiative effects that are not directly related to drifting-snow particles. The question of the role of drifting snow on cloud formation cannot be supported here by enough observations

and requires further investigations. Thus, we rejected all the cases where the increase in concentrations of cloud droplets, rain and ice crystals were among the 10 % highest increases between MAR-DR and MAR-nDR simulations.

Secondly, snowfall occurrence must be removed. Precipitating particles during snowfall can be distinguished between particles reaching the ground (designed as $snowfall_{ground}$) and particles sublimating entirely during their falling through the atmosphere (designed as $snowfall_{virga}$). As MAR-DR mixes $snowfall_{ground}$, $snowfall_{virga}$ and eroded snow particles in the snow

particle ratio, we use the MAR-nDR simulation to discard $snowfall_{ground}$ and $snowfall_{virga}$ occurrences from both simulations. Once $snowfall_{ground}$ cases in MAR-nDR have been excluded, $snowfall_{virga}$ have been identified from the remaining cases in MAR-nDR (i.e. without $snowfall_{ground}$) as all profiles with a snow particle ratio above $q_{snow0.9}$ at any vertical level. $q_{snow0.9}$ is the 0.9 quantile of the snow particle ratio at the lowest vertical level. As a consequence, we rejected many snowfall occurrences, including drifting-snow mixed with snowfall occurrence and more generally many cloudy periods. Few single clear outliers

remain after the filtering process (drifting-snow layer height > 2000 m) and are discarded.

Finally 16,699 simulated atmospheric profiles were available for analysis. Those profiles, designated as "filtered conditions", hold valuable information about the direct radiative contribution of eroded snow particles without interfering with the radiative contribution of snowfall or newly formed atmospheric water species.

## 3   Results

The ability of MAR to reproduce the drifting-snow climate of Adelie Land has been extensively evaluated in Amory et al. (2021), in which a close agreement with observations is demonstrated for the SMB, drifting-snow mass transport and frequency up to the scale of the drifting-snow event. We refer to this study for further details on the model evaluation regarding drifting snow. In this section, we focus on the impact of drifting snow on the representation of surface and near-surface meteorological variables. This is achieved by comparing two sets of simulations, in which the drifting-snow scheme has been respectively

switched on (referred to as MAR-DR) and off (referred to as MAR-nDR). The results are compared over periods for which observations are also available, i.e. 2010–2018 for near-surface wind speed, air temperature and relative humidity and 2014-2018 for radiative fluxes. Except for the surface turbulent fluxes, all the other meteorological variables are observed at D17. Half-hourly variables extracted from the surface or the lowest model level (2 m) and the nearest grid cell to the observation location are used for comparison. Modifications in near-surface and surface variables are summarized in a Taylor Diagram

(Taylor, 2001) presented in supplementary material (Fig. S1).

### 3.1   A case study

We firstly focus on a strong drifting-snow event that occured over the 1-3 October 2017 period at D17 to understand the physical processes involved in important changes between MAR-DR and MAR-nDR simulations.

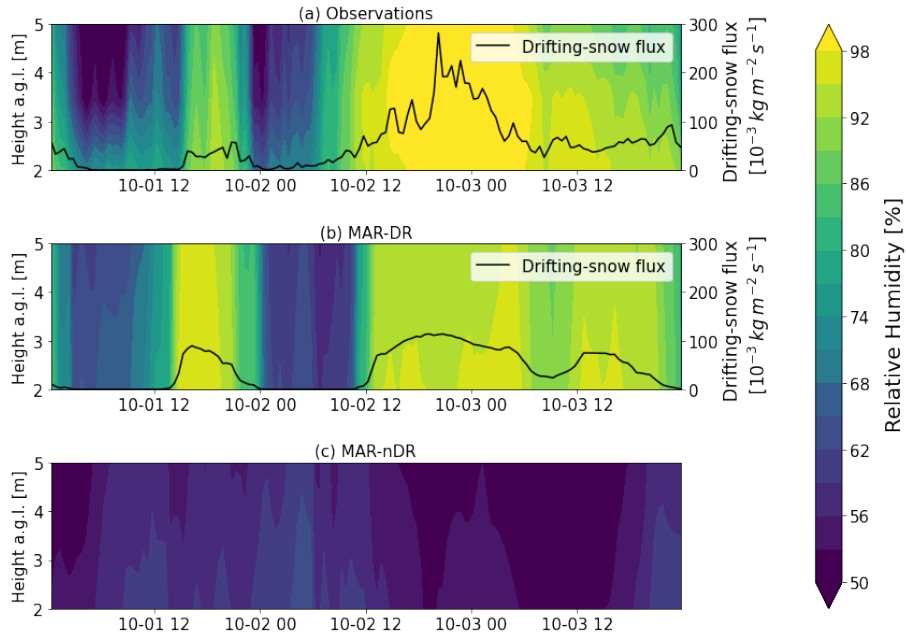

**Figure 2.** (a) Observed, (b) MAR-DR and (c) MAR-nDR vertical relative humidity profile (with respect to ice, color) and drifting-snow fluxes (from the surface to 2 m a.g.l., black line) between the 1st and the 3rd of October 2017.

### 3.1.1 Relative humidity, temperature and wind speed

MAR-DR captures the drifting-snow event in terms of timing and occurrence but underestimates the magnitude of the drifting-snow flux. The simulated drifting-snow flux is approximately three times lower than the observed flux at the peak of the event. The near-surface humidity budget is particularly impacted during this specific event (Fig. 2): when drifting snow occurs, a near-saturated layer develops in the lowest meters of the atmosphere. During the peak of the event, this layer reaches saturation. MAR-DR reproduces this observed increase in relative humidity, while relative humidity from MAR-nDR simulation is lower
by up to 47 %.

As no snowfall is simulated during this event, the high snow particle ratio found in the lower part of the drifting-snow layer can be largely attributed to snow eroded from the surface by the wind (Fig. 3 (a)). Once suspended in the atmosphere, those particles sublimate in proportion to the undersaturation of ambient air (Schmidt, 1982) (Fig. 3 (b)), and relative humidity increases due to moisture release and consumption of latent heat and subsequent cooling of the atmosphere. This cooling
decreases the 2 m temperatures (Fig. 3 (f)), also reducing the positive temperature bias in comparison to in situ observations. Wind speeds increase in MAR-DR compared to MAR-nDR (Fig. 3 (h)), which are discussed in more detail in Sect. 4.1. Figure S6 illustrates the impact of drifting snow on vertical profiles of temperature, relative humidity and wind speed in the model at D17 during this event.

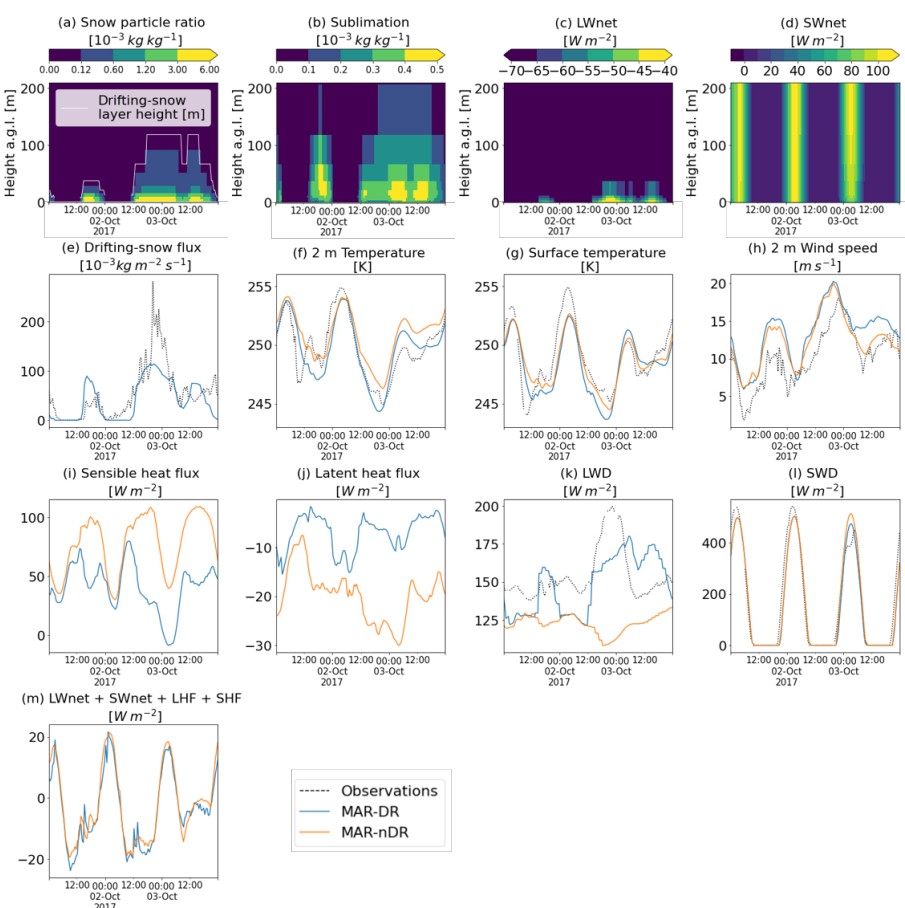

**Figure 3.** (a) Snow particle ratio, (b) sublimation (expressed in g of sublimated snow per kg of moist air per 30 minutes), (c) SWnet and (d) LWnet vertical profiles as simulated by MAR-DR during a drifting-snow episode occurring between the 1st and the 3rd of October 2017. (e) to (m): 2 m and surface variables as observed and simulated by MAR-DR and MAR-nDR between the 1st and the 3rd of October 2017.

### 3.1.2 Incoming radiative fluxes, turbulent fluxes and surface energy budget

During the 1-3 October 2017 period, both radiative and turbulent fluxes at the surface were modified by drifting snow. On one hand, in MAR-DR during modelled drifting-snow occurrences, the net longwave radiation (LWnet), defined as LWD+LWU and calculated at each model vertical level, increases close to the surface and peaks where the snow particle ratio is maximum (Fig. 3 (c)). Conversely, the net shortwave radiation (SWnet), defined as SWD+SWU, decreases (Fig. 3 (d)). Those modifications are transferred towards the surface where large differences in LWD and SWD between MAR-DR and MAR-nDR are visible

(Fig. 3 (k), (l)), suggesting the drifting-snow layer acts as a near-surface cloud by enhancing LWD and decreasing SWD at the surface. Yamanouchi and Kawaguchi (1984) observed similar LWD and SWD variations over the first 30 m of the atmosphere at Mizuho station during drifting-snow episodes.

On the other hand, surface turbulent fluxes (not observed, see Sect. 2.1) is reduced when drifting snow is considered in the model (Fig. 3 (i), (j)). The development of a near-saturated layer over the first meters of the atmosphere due to drifting-snow sublimation (Fig. 2) reduces the humidity gradient and prevents surface sublimation with less latent heat exchange at the surface. Furthermore, atmospheric sublimation cools the atmosphere, inducing reduced vertical temperature gradients and lower sensible heat fluxes at the surface.

As a summary, we find that drifting snow induces an increase in net radiative fluxes ($+22.5 W m^{-2}$ on average during the considered period), which is driven by increasing LWD and decreasing SWD. This compensates for modifications in turbulent fluxes ($-24.06\ W m^{-2}$). Consequently, our simulations suggest that notable impacts on the energy inputs result in a negligible change in the final energy budget and surface temperature during this specific period.

## 3.2 Seasonal modifications

Drifting snow not only impacts the near-surface meteorology during specific events, but it also modifies their seasonal cycle at D17 (Fig. 4). As drifting snow becomes more frequent in winter (March to October) in Adelie Land due to the increased katabatic forcing (Amory, 2020), its related impacts on the lower atmosphere are most notable during that period. As a result, relative humidity and LWD biases are notably reduced in winter in MAR-DR compared to MAR-nDR.

### 3.2.1 Relative humidity, temperature and wind speed

At 2 m above ground level, the observations highlight a seasonal cycle in relative humidity (Fig. 4, (a)). MAR-DR captures this seasonal cycle, whereas MAR-nDR simulates nearly constant monthly means of relative humidity. As drifting-snow frequency and mass transport increase in winter Amory (2020), more airborne snow particles become available for sublimation. The lower temperatures in winter together with the additional atmospheric cooling and moistening caused by drifting-snow sublimation result in an increase in near-surface relative humidity. This sometimes leads to the establishment of a nearly saturated air layer over several meters (Fig. 2). Sublimation of airborne snow particles is responsible for a 13.9 % mean increase in 2 m relative humidity at D17 (2010–2018), which is consistent with previous simulations in this area (Lenaerts and van den Broeke, 2012). Taking into account drifting snow notably lowers the relative humidity root mean squared error (RMSE) by 39 % (Table 2), suggesting that drifting-snow sublimation mainly governs temporal variations in relative humidity at D17 in agreement with Amory and Kittel (2019).

Drifting snow also accounts for lower 2 m temperatures in MAR-DR at D17 (-0.7 K on average), particularly during winter (Fig. 4, (c)). However, further analysis shows that the most important temperature modifications occur higher up in the drifting-snow layer (Sect. 4.1). Accounting for drifting snow in MAR reduces the 2 m temperature positive bias in comparison with observations (divided by a factor 2.5, Table 2).

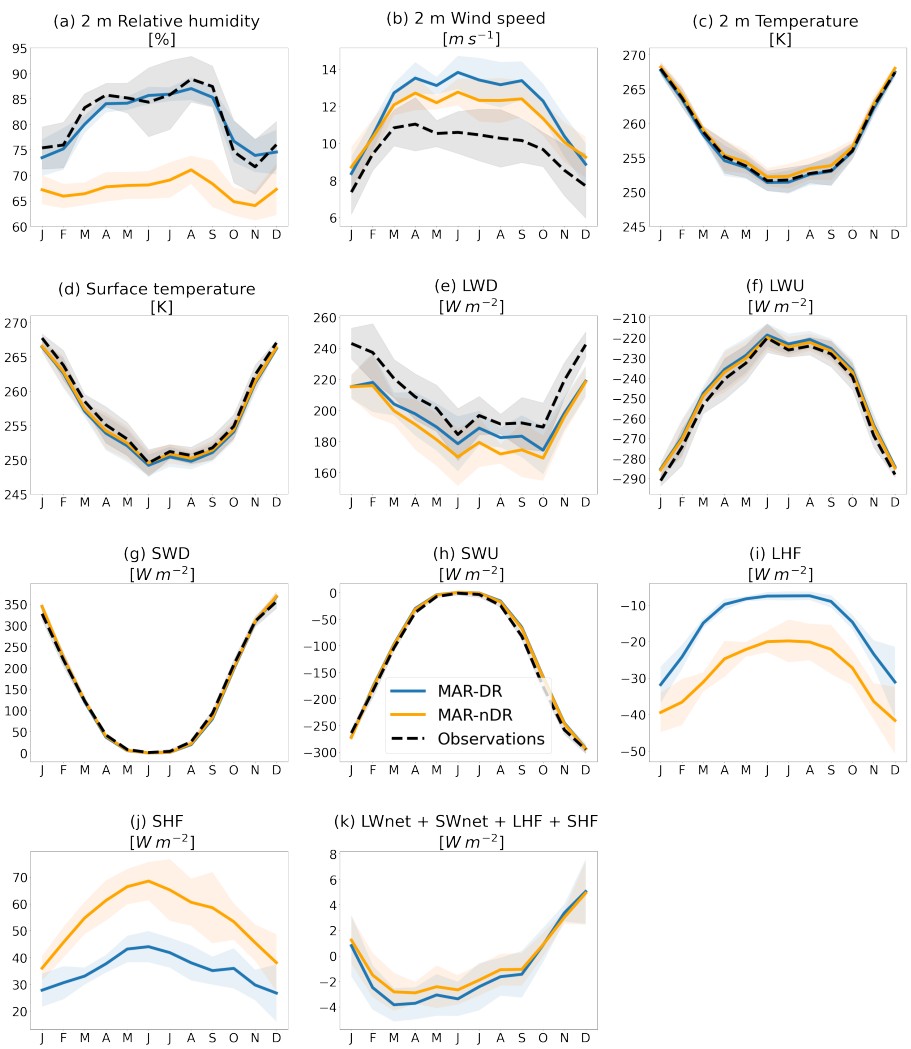

**Figure 4.** 2 m and near-surface variable monthly means as simulated by MAR-DR and MAR-nDR. First, data are aggregated by both months and years. Then means and standard deviations are evaluated within each group aggregated by month. Statistics are performed on 2014–2018 period for radiative fluxes and surface temperature and on 2010–2018 period for near-surface variables and turbulent fluxes.

**Table 2.** Root mean square error (RMSE), Pearson correlation coefficient and mean bias computed at D17 for MAR-DR and MAR-nDR half-hourly simulations in comparison with in situ observations.

| | r | | RMSE | | Mean bias | |
|---|---|---|---|---|---|---|
| | MAR-DR | MAR-nDR | MAR-DR | MAR-nDR | MAR-DR | MAR-nDR |
| LWD [$Wm^{-2}$] | 0.87 | 0.89 | 19.9 | 22.8 | -14.9 | -20.4 |
| LWU [$Wm^{-2}$] | 0.97 | 0.98 | 6.5 | 5.6 | -4.0 | -2.9 |
| SWD [$Wm^{-2}$] | 0.98 | 0.98 | 24.6 | 24.2 | -1.3 | 0.3 |
| SWU [$Wm^{-2}$] | 0.98 | 0.98 | 22.4 | 22.0 | -7.0 | -5.9 |
| Surface temperature [K] | 0.97 | 0.98 | 1.7 | 1.4 | -1.0 | -0.7 |
| 2 m temperature [K] | 0.97 | 0.98 | 1.3 | 1.2 | -0.2 | 0.5 |
| 2 m wind speed [] | 0.78 | 0.82 | 3.0 | 2.5 | 2.3 | 1.7 |
| 2 m relative humidity [%] | 0.62 | 0.51 | 9.5 | 15.8 | -0.7 | -14.0 |

### 3.2.2 Incoming radiative fluxes, turbulent fluxes and surface energy budget

Drifting snow enhances the seasonal values of LWD (Fig. 4 (e)), but even if significant modifications in SWD can occur during specific events such as presented in Fig. 3 (d), the impact on seasonal averages is low (Fig. 4 (g)). LWD are mainly modified during winter, in pace with the seasonal cycle of drifting snow. LWD modifications are even more visible during this period because atmospheric temperatures reach their minimum. As MAR-nDR underestimated LWD at D17 (the mean bias equals to -20.4 $Wm^{-2}$), the LWD negative bias is reduced to -14.9 $Wm^{-2}$ in MAR-DR, showing that the latter simulates more realistic LWD values in winter. The impact of drifting snow on other incoming and outgoing radiative fluxes at the surface is lower (Table 2).

Drifting snow accounts for a significant decrease in SHF and LHF, i.e. larger than the interannual variability (taken as the standard deviation computed from annual means) during the 2010–2018 period. By increasing relative humidity in the lower atmosphere (Fig. 4 (a)), drifting-snow sublimation decreases the vertical gradient in humidity and limits latent heat exchanges with the surface. As a consequence, surface sublimation is locally reduced by a factor of 2.0 in MAR-DR (LHF, expressed in W m-2, Fig. 4(i)). However, at the same time, drifting-snow sublimation cools the boundary layer, reduces vertical temperature gradients and counterbalances the decrease in LHF by a decrease in SHF (Fig. 4 (j) (SHF divided by a factor 1.9 at D17).

Overall, drifting snow shows very little impact on the energy budget (LWnet + SWnet + LHF + SHF = -0.5 $Wm^{-2}$). This results from a compensation between the net drifting-snow radiative forcing (difference in SWnet + LWnet between simulations = 6.1 $Wm^{-2}$) and the surface turbulent heat fluxes (difference in LHF + SHF between simulations = -6.6 $Wm^{-2}$). As the energy available at the surface has not been notably modified by introducing drifting snow in MAR-DR, surface temperature is almost unchanged (-0.3 K, Fig. 4 (d)).

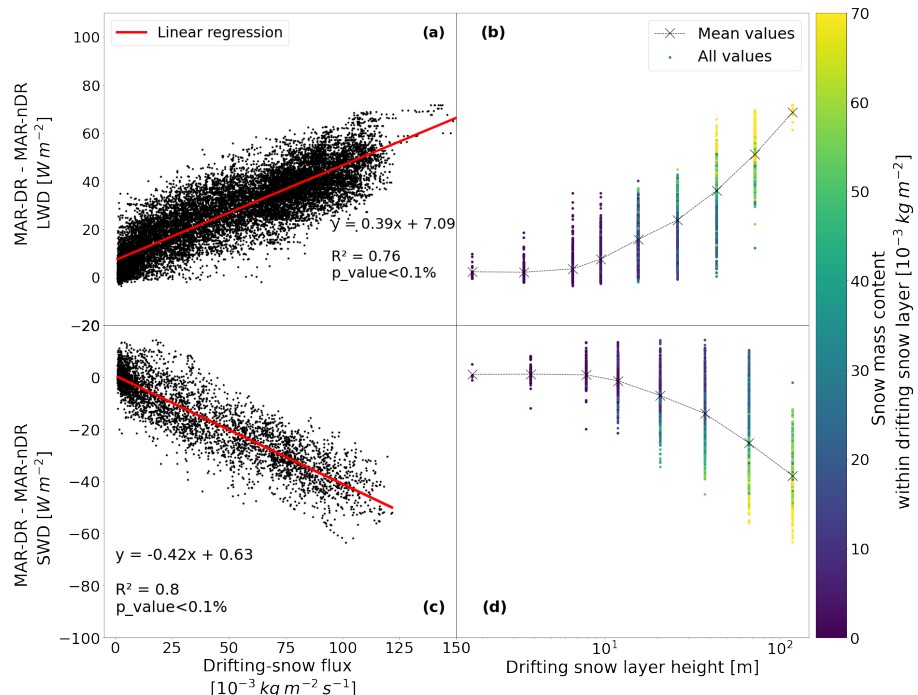

**Figure 5.** Modifications in (a) LWD and (c) SWD between MAR-DR and MAR-nDR during drifting snow (drifting-snow flux > $10^{-3}\,kg\,m^2\,s^{-1}$), as a function of drifting-snow flux, for a mean flux calculated between 0 and 2 m. The red line indicates the best linear regression between radiative modifications and drifting-snow fluxes. Regression functions and statistics are displayed on the corresponding panels. SWD modifications are computed when MAR-nDR simulates SWD > 50 $Wm^{-2}$. Data are filtered according to Sect. 2.4.
Modifications in (b) LWD and (d) SWD between MAR-DR and MAR-nDR during drifting snow (drifting-snow flux >$10^{-3}\,kg\,m^2\,s^{-1}$), as a function of drifting-snow layer height. The colorbar indicates the mass of snow contained between the drifting-snow layer height and the surface. Mean values are calculated for the lowest 9 model vertical levels and are represented by a grey mark. SWD modifications are computed when MAR-nDR simulates SWD > 50 $Wm^{-2}$. Data are filtered according to Sect. 2.4.

## 3.3 Impact on incoming radiation

In this section the impact of drifting snow on incoming radiation is analysed. In order to more specifically focus on the radiative contribution of eroded snow particles in MAR-DR, we filtered data according to Sect. 2.4 . Thus, we discarded cases with snowfall or modifications in the cloud structure between MAR-DR and MAR-nDR.

We observed that under such conditions, LWD modifications correlate linearly with the drifting-snow flux. Conversely, when significant SWD reaches the surface (> 50 $Wm^{-2}$), SWD decreases linearly with the drifting-snow flux (Fig. 5 (a), (c)). Furthermore, large modifications in incoming radiative fluxes are associated with thick and dense drifting-snow layers (Fig. 5 (b), (d)).

According to Fig. 5, the most significant drifting-snow events (drifting-snow fluxes $>= 75\,10^{-3}\,kg\,m^2\,s^{-1}$ and layer height $> 100$ m) can lead to large increases in LWD: the largest increase occurs in June 2013 with $+ 72\ Wm^{-2}$. This effect is on average partially compensated by SWD decreases: the most notable decrease in SWD is reached in September 2010 with -63 $Wm^{-2}$. As suggested by our simulations, the net drifting-snow radiative forcing is positive ($+ 6.1\ Wm^{-2}$, mean value on the all unfiltered dataset), particularly during filtered conditions ($+ 24.5\ Wm^{-2}$), even when SWD are significant ($+ 12.8\ Wm^{-2}$). The effect is more prominent during low solar irradiance periods because LWD is highly impacted whereas SWD is absent (or very low) and cannot be modified. The additional radiative forcing due to drifting snow is higher when eroded snow particles predominantly contribute to the suspended snow mass.

Snow erosion and resulting drifting-snow sublimation modify the atmospheric composition in water species by introducing additional snow particles in the atmosphere and also enhancing its water vapor content, resulting in an increase in longwave emissivity (Yamanouchi and Kawaguchi, 1984). A sensitivity analysis was performed in order to distinct and quantify the relative contribution of eroded snow particles and additional water vapor to modified radiative fluxes. In addition to MAR-DR and MAR-nDR, two other runs with and without drifting snow were performed for the year 2017 in which the radiative contribution of snow particles has been disabled. The difference between the two last runs were compared to differences between MAR-DR and MAR-nDR and demonstrated that incoming longwave modifications are predominantly due to the radiative contribution of drifting-snow particles (Fig. S2).

## 4   Discussion

### 4.1   Impact on the boundary layer

In the model at D17, the boundary layer is predominantly impacted on the first 600 m a.g.l. (lowest 11 vertical levels, Fig. 6). The snow particle ratio is high near the surface where snow erosion occurs, and decreases rapidly with height above the surface (Fig. 6 (j)). However, atmospheric sublimation peaks higher up (fourth model vertical level, 12 m), as already suggested in e.g. van den Broeke et al. (2006) and Amory and Kittel (2019). As drifting-snow sublimation is a self-limiting process inhibited by the development of near-saturated layers close the the surface (Bintanja, 2001), the maximum in atmospheric sublimation during drifting-snow events occurs higher up in the atmosphere where sublimation is favoured by the under-saturation of the environment.

As observed by Palm et al. (2018) using dropsondes across Antarctica, well-mixed layers with small vertical gradients in temperature and increasing relative humidity with the proximity of the surface characterize the thermodynamic structure of drifting-snow layers. Figure 6 shows that both features are reproduced by MAR, with a well-mixed temperature structure ($< 0.019\ Km^{-1}$) within the first 100 m above ground, i.e. near the average height of drifting-snow layers, and a downward positive gradient in relative humidity. Further evaluation is however necessary to quantify the ability of the model to capture wind-shear-induced turbulent mixing and warm-air entrainment within katabatic flows.

According to the vertical profiles in Fig. 6, increases in wind speed, relative humidity, specific humidity and decreases in temperature are in accordance with increases in atmospheric sublimation. All of these variables are predominantly modified in

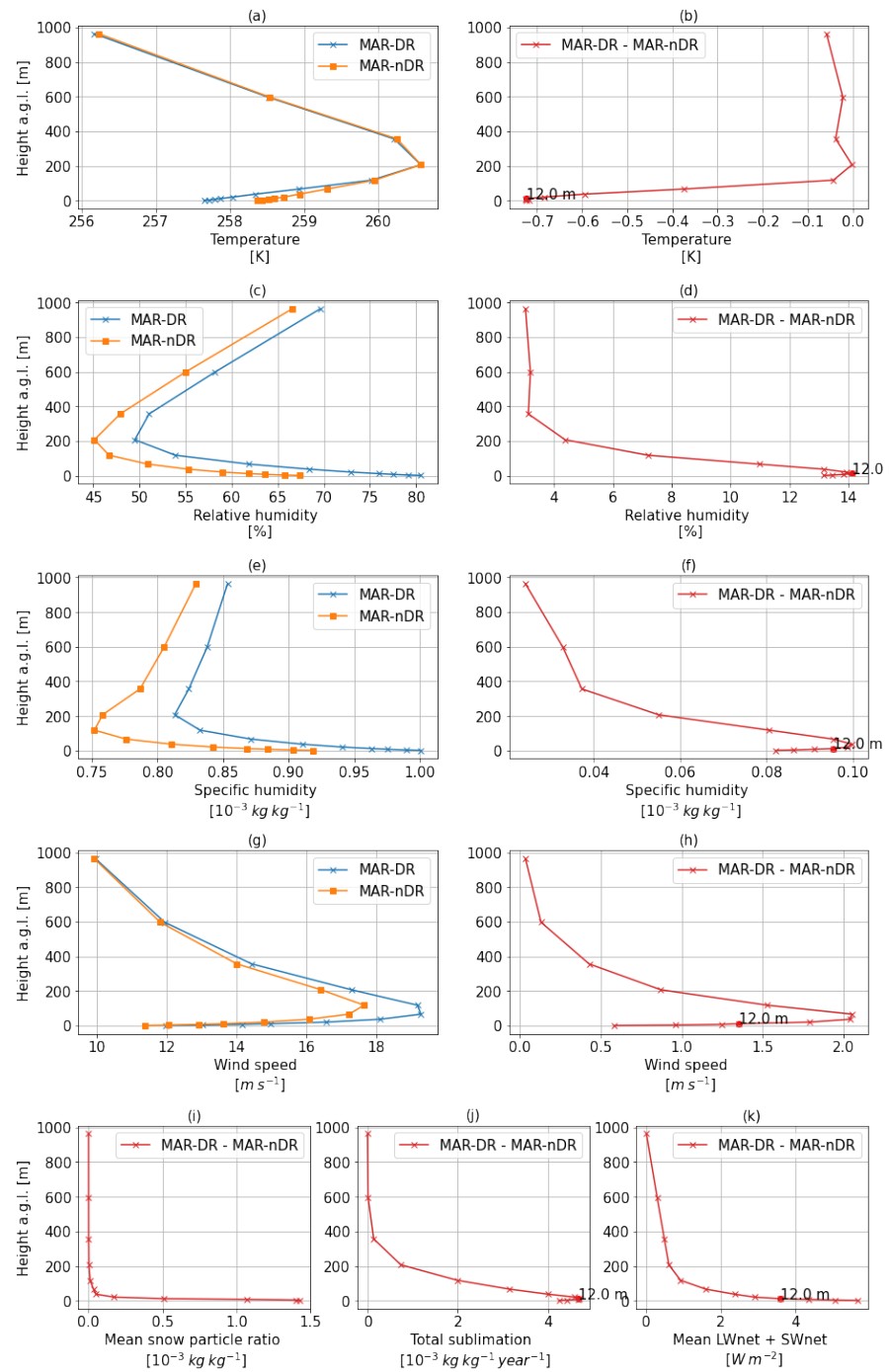

**Figure 6.** Annual mean (2010-2018) vertical profiles for near-surface and surface variables calculated at D17 on the lowest 12 vertical levels as simulated by MAR-DR, MAR-nDR or corresponding differences between both runs. In (i), sublimation rates are expressed in g of sublimated snow per kg of moist air per year.

the drifting-snow layer when accounting for drifting snow. The modification intensity peaks in the vicinity of the vertical level experiencing maximum atmospheric sublimation. This suggests that atmospheric sublimation drives the impacts of drifting snow on the low-atmosphere meteorology at D17. Accounting for this phenomenon at each vertical level modifies the entire

boundary-layer structure.

Wind speed increases in MAR-DR compared to MAR-nDR at D17 (Fig. 6 (g), (h)). The largest increases are found at the sixth and seventh vertical levels (38 m and 67 m), near the level experiencing maximum sublimation (fourth vertical model level, 12 m). As already suggested (e.g., Kodama et al., 1985), wind speed can increase during drifting snow events because of increased density of the air-snow mixture and an increased stable thermal stratification (Fig. 6 (a)) caused by the atmospheric

sublimation-induced cooling, which is a positive feedback on a sloping surface due to the gravitational nature of katabatic winds (Bintanja, 2000). This effect could be moderated at the lowest model vertical levels by surface-atmosphere interactions, such as the surface drag responsible for a decrease in wind speed. Further analysis reveals that incorporating snow particles in the calculation of the virtual potential temperature, in order to simulate the contribution of snow particles to the air density has a small impact on wind speed in MAR-DR (Fig. S4). Finally, a supplementary analysis (Fig. S3) on the sensitivity of the

katabatic forcing term to the inclusion of drifting snow is proposed through a computation of the potential temperature deficit in the low atmosphere at D17, following Van den Broeke and Van Lipzig (2003). Decreasing temperatures with increasing drifting-snow sublimation modify mean potential temperature in the boundary layer. Such modifications are responsible for an increase in katabatic forcing in MAR-DR. Moreover, higher wind speeds have the potential to (i) erode more snow, (ii) advect drifting-snow particles at higher elevations in a warmer and drier environment through turbulent mixing, (iii) increase

the residence time of drifting-snow particles in the atmosphere. Consequently, higher wind speeds trigger three factors that could potentially favor more atmospheric sublimation and constitute a positive feedback. We explore this feedback in Fig. 7 (a) where MAR-DR drifting-snow fluxes are computed at each model vertical level and shown as monthly averages. Additionally, we performed the same computation by replacing the wind speed with the one from the simulation MAR-nDR, which is on average lower than in the MAR-DR simulation. We aim here at quantifying the absence of the positive feedback of sublimation

on wind speed and its impact on drifting-snow fluxes. Figure R3 shows that stronger wind speeds reinforced by additional sublimation in the MAR-DR simulation are responsible for an increase in drifting-snow fluxes. Such drifting-snow fluxes are correlated with atmospheric sublimation in a logarithmic fashion (Fig. R3 (b)): higher wind speeds induce higher drifting-snow fluxes which are in turn responsible for enhanced atmospheric sublimation. Enhancement of sublimation is however limited by the self-limiting feedback of sublimation (Bintanja, 2001), the latter being dependent on the under-saturation of the ambient

environment (see colorbar on Fig. R3 (b)). Ultimately, our simulations suggest that the feedback of increased wind speed on atmospheric sublimation could be all the more important at higher elevations (ex: sixth model vertical level, 38 m) where the atmospheric sublimation potential is more sensitive to increases in drifting-snow fluxes due to a lower relative humidity.

Finally, we find that the net radiative budget (LWnet + SWnet) in the model vertical levels increases with the proximity of the surface within the drifting-snow layer (Fig. 6 (k )). This increase is due to suspended snow particles in the drifting-snow

layer, which emit longwave radiation and trap heat, consequently inducing a warming effect that competes with cooling by

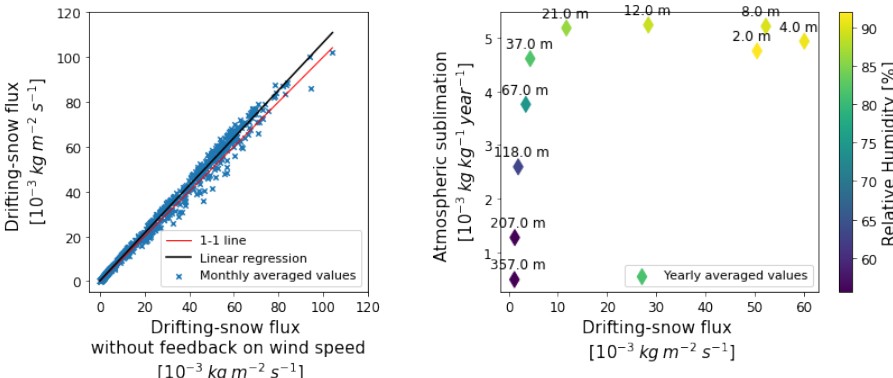

**Figure 7.** (a) Comparison between drifting-snow fluxes in the atmosphere calculated in MAR-DR using usual wind speed values (y axis), or using wind speed values retrieved from the MAR-nDR simulation (x axis). The latter is done to approximate drifting-snow fluxes without accounting for the impact of drifting-snow sublimation on wind speeds. All fluxes are monthly averaged values over the period 2010-2018, computed at each of the 10 lowest model vertical levels. The black line denotes the best linear regression. Taking into account the atmospheric sublimation feedback on wind speed enhances drifting-snow fluxes. (b) Atmospheric sublimation as a function of drifting-snow fluxes for the 10 lowest model vertical levels. Values are averaged yearly to denote the model vertical level elevation (black text). Annual atmospheric sublimation rates are expressed in kg of sublimated snow mass per kg of moist air. The colorbar indicates yearly averaged relative humidity at the considered level. Enhanced drifting-snow fluxes are responsible for increased atmospheric sublimation, until a plateau is reached. This plateau coincides with the occurrence of near-saturated environments, where additional sublimation is limited by the negative feedback of sublimation.

sublimation. The comparison between MAR-DR and MAR-nDR indicates that the net effect is a decrease in atmospheric temperatures (Fig. 6 (a) and (b)), so the model suggests that the cooling effect due to sublimation dominates.

## 4.2 Spatial analysis

By analysing our simulation at a regional scale, we demonstrate that the results obtained at D17 remain consistent at the scale
of the integration domain (Table 3).

Firstly, we estimate the drifting-snow magnitude at a regional scale by studying snow mass transport (Fig. 8 (a)). We calculate the mass of snow transported at the lowest atmospheric level at each grid point every year, and then calculate the annual mean values at each grid point. The spatial distribution of snow mass transport is closely related with wind speed (Amory et al., 2021). It is particularly enhanced where the topographic slope accentuates and favours channeling of katabatic winds, initiated
in the upper plateau region where wind speed and erosion are low.

The main drift-induced modifications in surface and near-surface variables described at D17 remain consistent at the scale of the integration domain. The spatial patterns of the differences in surface and near-surface variables correlate with spatial patterns of the snow mass transport as demonstrated by high and significant (p value < 0.01) Pearson correlation coefficients (Fig. 8). Modifications added by the drifting-snow scheme are often larger than the interannual variability (Fig. 8, undotted areas).

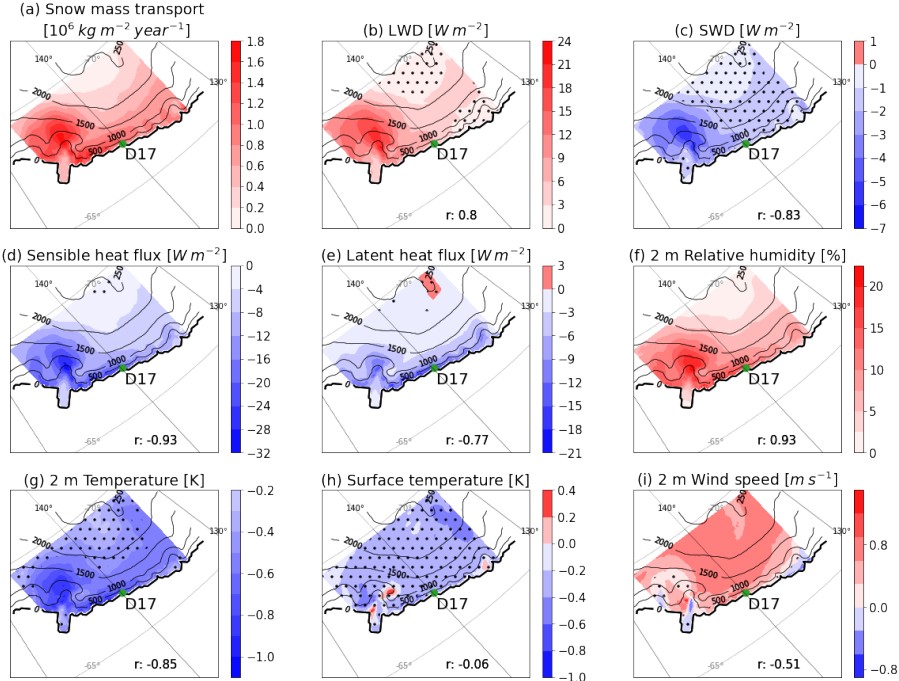

**Figure 8.** Annual mean (2010-2018) near-surface and surface variable modifications between MAR-DR and MAR-nDR over the integration domain. Within each panel, r indicates the Pearson correlation coefficient between the snow mass transport anomaly (a) and the considered variable (b to i). Dotted area designate areas where modifications are lower than interannual variability (taken as the standard deviation computed from annual means).

However, surface temperature and melt modifications (not shown) are poorly linked with snow mass transport (respectively r=-0.06 and r=0.3), another evidence that drifting snow does not modify the SEB significantly in the model.

    We estimate the net drifting-snow radiative forcing on Adelie Land to be + 7.6 $Wm^{-2}$ and the impact on SEB (estimated by LWnet + SWnet + LHF + SHF) to be smaller than 0.03 $Wm^{-2}$. Further spatial analysis performed at different vertical levels, and shown in the supplementary material (Fig. S2), indicate that the drifting-snow impacts within the boundary layer simulated 415  at D17 are also retrieved at a regional scale.

## 4.3   Current limitations

The validity of our results is affected, among others, by uncertainties related to the absence of model evaluation concerning surface turbulent fluxes and vertical profiles (Fig. 6), the scarcity of radiative measurements in Adelie Land and the current state of development of the model.

The vertical profiles presented in Fig. 6 have only been evaluated at 2 m. a.g.l. therefore the behavior of the model and the potential benefit of accounting for drifting snow in order to capture more realistic atmospheric dynamics in the lower

**Table 3.** Half-hourly mean value and standard deviation (STD) for several near-surface and surface meteorological variables computed on Adelie Land with MAR-DR and MAR-nDR. Differences between both model runs are attributed to drifting-snow processes.

| | MAR-DR | | MAR-nDR | | \|MAR-DR\| - \|MAR-nDR\| | |
|---|---|---|---|---|---|---|
| | Mean value | STD | Mean value | STD | Mean value | STD |
| LWD [$Wm^{-2}$] | 162.4 | 22.1 | 156.3 | 20.8 | 6.1 | 1.3 |
| LWU [$Wm^{-2}$] | -205.3 | 22.2 | -206.2 | 22.2 | -0.9 | 0 |
| SWD [$Wm^{-2}$] | 142.6 | 3.4 | 144.4 | 2.8 | -1.8 | 0.6 |
| SWU [$Wm^{-2}$] | -115.4 | 2.8 | -116.0 | 2.4 | -0.6 | 0.4 |
| LHF [$Wm^{-2}$] | -2.7 | 3.0 | -5.8 | 6.2 | -3.1 | -3.2 |
| SHF [$Wm^{-2}$] | 18.1 | 4.3 | 27.0 | 8.9 | -8.9 | -4.6 |
| LWnet + SWnet + LHF + SHF [$Wm^{-2}$] | -0.3 | 0.3 | -0.2 | 0.3 | 0.1 | 0 |
| Surface temperature [K] | 244.6 | 6.7 | 244.9 | 6.7 | -0.3 | 0 |
| 2 m temperature [K] | 245.5 | 6.7 | 246.0 | 6.8 | -0.5 | -0.1 |
| 2 m wind speed [$m\,s^{-1}$] | 10.6 | 1.8 | 10.1 | 1.8 | 0.5 | 0 |
| 2 m relative humidity [%] | 92.1 | 3.2 | 84.9 | 6.2 | 7.2 | 3.0 |

atmosphere still needs to be assessed. Daily radiosoundings are operated at the closeby permanent station of Dumont D'Urville. However, sufficient climatic disparity exists between D17 location, situated on the marginal slope of the Antarctic continent and Dumont d'Urville station, situated beyond the continent boundaries on an island approximately 15 kilometers northeast of D17. Nevertheless a good agreement with observed values for several meteorological variables (wind speed, relative humidity, temperature, drifting-snow flux, incoming and outgoing radiative fluxes) and the fact that the model is well constrained at its boundaries by global reanalysis is an argument in favour of firstly studying model outputs at the lowest vertical level and then exploring its behavior at higher altitudes. Dropsondes observations near D17 location or operation of radiosoundings from the ground at D17 would help assess model performance and uncertainties at higher elevation in complement of near-surface observations.

Modifications in downwelling atmospheric radiation, induced by the inclusion of drifting snow in MAR, are consistent with former in situ estimates (Lesins et al., 2009; Mahesh et al., 2003; Yamanouchi and Kawaguchi, 1984; Yang et al., 2014) of the radiative contribution of suspended particles, suggesting the model simulates a realistic radiative forcing. However, our results might be affected, among other, by limitations in the vertical resolution of the model which does not take into account the large variations of snow particles concentration over the first 2 m of the low troposphere, and the limitations of the current radiative scheme (e.g., Delhasse et al., 2020), inherited from the ERA-40 reanalysis product (Uppala et al., 2005). Improved/regressed evaluation statistics when accounting for drifting snow can be linked with error compensation elsewhere in the model, independently from the ability of the model to accurately reproduce drifting-snow processes. Radiation measurements are scarce in Antarctica due to the harsh environmental conditions and the difficulty to deploy and maintain measurement sites in remote areas, thus more in situ observations of radiative fluxes and drifting-snow layer properties are

needed for a more in-depth evaluation of model results and, in our case, assessment of the temporal and spatial representativity of the interactions described at site D17.

Drifting-snow sublimation, defined here by the difference in atmospheric sublimation between MAR-DR and MAR-nDR over the first 1000 m above ground, equals on average 606 mm we/year on the all Adelie Land domain, with higher values reported at D17 (716 mm we/year). These rates are larger than previous in situ estimates of drifting-snow sublimation held in distinct parts of the continent (King et al., 1996, 2001) where the climate differs from the windy and (relatively warm) conditions of coastal Adelie Land. However, Palm et al. (2017), through remotely sensed data, and Lenaerts and van den Broeke (2012) by using a regional climate model, report sublimation rates in Adelie Land which are more in agreement with our model estimates, though still twice to three times lower.

Finally, the inclusion of newly formed clouds in MAR-DR as discussed in Sect. 2.4 can contribute to the probable overestimation of drifting-snow sublimation rates in MAR-DR. The sublimation rates, as simulated by MAR, have not been yet directly compared to in situ measurements, although indirect comparisons have been made through the evaluation of near-surface air relative humidity and temperature. Accounting for drifting-snow sublimation in the present study has proven useful to modify the relative humidity of the lower atmosphere and help the model matching with observed relative humidity from a timescale of a single event (Fig. 2) to a seasonal scale (Fig. 3). The deployment of eddy-covariance systems including highly sensitive hygrometers could provide complementary atmospheric sublimation estimates to evaluate model simulations during calm to moderate conditions. However, using eddy-covariance devices during strong drifting-snow episodes remains a challenge as drifting-snow particles alter the observed signal and limit their use in Adelie Land (e.g., Bintanja and Reijmer, 2001). Moreover, including drifting snow in MAR shows large impacts on turbulent fluxes which compensate (and sometimes slightly override, e.g. at D17) modifications in radiative fluxes. Such a compensation also needs to be evaluated through comparison with direct in situ measurements of latent and sensible heat fluxes during drifting-snow occurrences to determine if MAR-DR simulates (more) realistic turbulent heat exchanges at the surface. Modelling hypothesis regarding drifting-snow particle distribution and subsequent sublimation rates could be better constrained using information derived from in situ optical measurements (e.g., Naaim-Bouvet et al., 2013).

Additionally, we introduced a method, based on CALIPSO observations to estimate the height of a drifting-snow layer using model outputs. This method allows us to derive an objective criterion concerning snow concentration in the atmosphere to determine the presence (or not) of a drifting-snow layer and its height, during specific meteorological conditions. This method is limited by the fact that it has only been developed for 8 years of CALIPSO observations collected near D17. Future work could focus on other locations in Antarctica to improve the determination of the snow concentration threshold by gathering more remotely sensed observations to be compared to model simulations. This could ultimately lead to an evaluation of modelled drifting-snow layer heights using CALIPSO observations on a specific test dataset. Ultimately, the use of a grounded lidar at D17 could provide complementary information concerning the vertical structure of drifting-snow layers.

Finally, we underline that independent modelling approaches lead sometimes to contrasted results, highlighting the uncertainty related to modelling choices. For example, energy exchange following atmospheric sublimation can be accounted for in the SEB (Lenaerts and van den Broeke, 2012) or computed at every model vertical level (this study) and ultimately lead

to distinct impacts of drifting snow on the simulated climate. Intercomparing drifting-snow models and related drifting-snow processes could be of a great interest for the Antarctica's regional modelling community, and such a work would require simulations performed under comparable conditions (e.g. same region, comparable horizontal and vertical resolution, same boundary forcing).

## 480 5 Summary and conclusion

We investigated the impact of drifting snow on the low atmosphere and the surface in coastal Adélie Land by comparing two simulations, respectively with and without drifting snow, performed with the latest version of the regional climate model MAR (MARv3.11) over a 9-year-long period. Simulating drifting snow leads to notable modifications in near-surface and surface variables. Our results suggest such effects are mainly driven by additional sublimation of drifting-snow particles in the low-
485 level atmosphere. Temperature decreases (- 0.5 K on average, -0.7 K at D17) and relative humidity increases at 2 m a.g.l. (+7.2 % on average, +13.3 % at D17) when drifting snow is taken into account, as a result of the latent heat exchanges and the release of additional water vapor. Modifications in temperature and relative humidity are not largest at the surface where snow mass transport is the most intense, but peak higher in the drifting-snow layer at the fourth (12 m) atmospheric level in agreement with the magnitude of atmospheric sublimation.

Wind speed increases in MAR-DR compared to MAR-nDR on the integration domain and at D17 (Fig. 6 (g), (h)). At D17, the largest increases are found at the sixth and seventh vertical levels (38 m and 67 m), near the level experiencing maximum sublimation (fourth vertical model level, 12 m). Thus, we observe a strong influence of drifting-snow sublimation on the structure of the boundary layer in the model, highlighting the importance of computing latent heat exchanges at each vertical level. When compared to in situ data observed at D17, 2 m relative humidity representation is greatly improved. The RMSE is
reduced from 15.8 % to 9.5 % and the mean bias is reduced from -14.0 % to -0.7 %. Additionally, the 2 m temperature mean bias is also reduced (RMSE equals 1.3 K and 1.2 K respectively, the mean bias is reduced from 0.5 K to -0.2 K).

We observe significant modifications in radiative and turbulent components of the SEB when taking into account drifting snow. The presence of a drifting-snow layer leads to modifications similar to the presence of a near-surface cloud, inducing enhanced LWD and decreased SWD. When the simulations are neither affected by snowfall nor drift-induced modifications in
cloud structure (Sect. 2.4), we observed that the higher the drifting-snow flux or the thicker the drifting-snow layers, the greater the modifications in radiative fluxes. As a result, LWnet increases at the surface (+7.0 $Wm^{-2}$ on average), SWnet decreases (-1.2 $Wm^{-2}$ on average), and the net effect is a positive drifting-snow radiative forcing of +5.8 $Wm^{-2}$. This is however mostly compensated in the drifting-snow layer by drifting-snow sublimation and at the surface by changes in turbulent fluxes. Atmospheric sublimation cools the lower atmosphere and reduces temperature and humidity gradients between the surface
and the atmosphere, inducing less latent heat consumed and less sensible heat provided at the surface. The net effect between modification in LHF and SHF is less energy being provided at the surface (-5.8 $Wm^{-2}$). Consequently, we observed negligible impact on energy supply at the surface and no significant impacts on surface temperature between simulations.

As a consequence, this study shows that differences in terms of surface temperature and heat budget are limited between MAR-DR and MAR-nDR simulation. However, impacts on each energy flux are significant; changes are compensating each other. Consequently, calibrating MAR with surface temperature data would likely lead to similar scores in Antarctica for simulations accounting / not accounting for drifting-snow processes. Nevertheless, drifting snow is a major component of both the surface mass balance and atmospheric moisture budget in the windy coastal area of Adelie Land (Amory and Kittel, 2019; Amory et al., 2021). Thus accurately accounting for drifting snow improves the ability of the model to capture the atmospheric thermodynamics and interactions with the surface in a current climate. As air moisture, LWD, SHF and LHF could very likely vary in a changing climate, capturing drifting-snow processes is consequently a key for higher confidence in climate and surface mass balance projections. Furthermore, drifting snow induces modifications in the snow isotopic composition. Additionally to snow redistribution which alters stratigraphy measurements, drifting-snow sublimation, as a major contributor to the air moisture budget, scrambles relationships between water stable isotopes composition and climatic variables (Bréant et al., 2019). Improving quantification of modelled drifting snow and related sublimation is a first step before (i) implementing isotopes in MAR and (ii) improving uncertainty assessment for climate reconstructions (Landais et al., 2017).

As illustrated in Sect. 4.1, drifting snow modifies the low-atmosphere structure and thermodynamics. In particular, larger moisture content and higher relative humidity in the drifting-snow layer reduces the capacity of the low-level atmosphere to sublimate snow particles during snowfall (Grazioli et al., 2017), potentially impacting modelled snowfall rates at the surface. By increasing atmospheric moisture, drifting snow can also impact cloud formation and physical properties. Lenaerts and van den Broeke (2012) reported increasing snowfall when accounting for drifting snow with RACMO2.1/ANT in peripheral regions of the ice sheet, including coastal Adelie Land. As a consequence, drifting snow may have additional impact on the SEB and SMB. These additional processes have not been investigated in our study, mainly because MAR-DR in its current version does not discriminate between eroded and cloud-originating snow particles. Drift-induced modifications in cloud structure and precipitation would benefit further investigation, and are left for future work. Separating eroded snow from snowfall particles in the model could enable prescription of different particle properties and pave the way for improvements in the representation of the drifting-snow radiative forcing and more generally in the representation of the drifting-snow impact on the low-atmosphere and the ice sheet surface.

*Data availability.*  Radiation data are available at Amory et al. (2020b). Meteorological and drifting-snow data are available at Amory et al. (2020a). MAR simulations are freely available by contacting the authors.

*Author contributions.*  L.L.T., C.A. and V.F. designed the study. C.A. ran the simulations. L.L.T post-processed data, and wrote the first draft. C. A., C. K., X. F. and H. G. developed the model. V. K. processed CALIPSO data. C.A. and V.F collected field data. All authors contributed to the manuscript and discussed the results.

*Competing interests.* The authors declare that they have no conflict of interests.

*Acknowledgements.* This work would not have been possible without the financial and logistical support of the French Polar Institute IPEV
(programme CALVA-1013 and GLACIOCLIM-SAMBA-411), and of the French Agence Nationale de la Recherche (projects ANR-14-
CE01-0001 (ASUMA) and ANR-16-CE01-0011 (EAIIST)). The authors thank all the on-site personnel in Dumont d'Urville and Cap
Prud'homme for their precious help in the field. Computational resources have been provided by the Consortium des Équipements de
Calcul Intensif (CÉCI), funded by the Fonds de la Recherche Scientifique de Belgique (F.R.S. – FNRS) under grant no. 2.5020.11, and the
Tier-1 supercomputer (Zenobe) of the Fédération Wallonie-Bruxelles infrastructure funded by the Walloon Region under grant agreement
no. 1117545.

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
