# Peer review of "Sensitivity of the surface energy budget to drifting snow as simulated by MAR in coastal Adelie Land, Antarctica"

_The Cryosphere, 2020_

## Referee Comment (RC1) · Anonymous Referee #1 · 8 Dec 2020

This paper examines the effects of snowdrift on Antarctica in a regional climate model with and without snowdrift physics included, and verified with local observations in Adelie Land. Overall it is an interesting paper with useful results, showing that the inclusion of snowdrift physics considerably reduces the bias with observations. I recommend publication provided that the following issues are satisfactorily addressed.

1. In places, the grammar is quite poor. For instance, there are many errors in the use of plural/singular. I strongly recommend to have the manuscript proofread and checked by a native speaker or the like.

2. In the model description section, I totally miss how snowdrift affects the momentum

balance of the boundary layer. I suspect that the enhanced drag of saltating particles is included (as it should be), but is the bouyancy effect of the suspending particles also included? I recommend adding information on how the model handles these physics, since the authors do show results on wind speeds, which are interesting (see below).

3. The effects on wind speed. These are quite interesting and this should be considerably expanded. With snowdrift, wind speed reduces at the surface (due to drag of saltating particles), but increases higher up (owing to stronger cooling associated with enhanced snowdrift sublimation, and the consequent reinforced katabatic forcing). However, a detailed analyses, including a 2D framework, is lacking. Now this is mentioned casually in two sentences (lines 345-347), but this topic warrants an in-depth discussion. One possibility is to include high-level winds in a panel of Fig. 7, and add a discussion on the snowdrift-induced effects of surface drag and katabatic forcing as mentioned above. Another option is to look at the momentum budget of the boundary-layer, and evaluate the friction and bouyancy/katabatic forcing terms. This will shed more light into the (spatial, vertical) variations in e.g. wind speed changes caused by the inclusion of snowdrift.

4. The changed momentum budget and wind speed feed back on the magnitude of snowdrift transport, saltation and suspension. In essence, the stronger winds in the boundary layer in the case of snowdrift will enhance snowdrift (sublimation, suspension), etc. This feedback is worth exploring, as it may vary in sign depending on distance to the surface; this model provides the perfect means to do so.

5. The comparison with other models should be greatly expanded. In section 4.3 there's a very brief discussion on RACMO, but this is insufficient. Readers would like to know in much more detail how your model results agree or differ from those obtained by RACMO, including the underlying physics. Otherwise this is "just" another model that studies the effect of snowdrift. Much can be learned by differences among models, especially about governing processes.

---

## Referee Comment (RC2) · Anonymous Referee #2 · 25 Jan 2021

This is an interesting study on the impact of drifting and blowing snow on boundary layer meteorology, surface radiation and energy balance in Terre Adélie, a region on the slopes of the East Antarctic ice sheet. A 9-yr time series (2010-18) of observations at a site near the coast is used to validate the regional climate model MAR. Methodology, presentation and discussion of results as well as the conclusions are sound. One of the main findings is that sublimation of drifting-snow particles leads at the surface to a reduction in sensible and latent heat exchange, which is compensated by an increase in net radiative forcing. While the net impact on total surface energy budget, and therefore surface temperature, is minimal, structure of the lower atmosphere is modified, which needs to be resolved in climate models to understand impact of warming on air-surface

interactions and boundary layer meteorology.

I have only minor comments, and recommend publication after they have been addressed.

1. It would be useful to get more detail on how sublimation rates are computed in MAR and how reliable they are. What are the model assumptions (snow particle size distribution and shape). Another model parameterisation of bulk sublimation rates from blowing snow by Déry and Yau (1999) uses the mean snow particle diameter. Is this the case in MAR and is particle diameter a sensitive parameter? If yes, future studies would gain by deploying next to an electro-acoustic sensor also an optical particle counter, to measure particle diameter as well as snow mass flux more accurately.

What are the uncertainties of calculated sublimation rates and how do calculations compare to existing observations in Antarctica (e.g. hourly blowing snow sublimation rates at Halley range 0.1-1 mm we/day (King et al., 1996))?

2. The limitations of the current evaluation method needs to be expanded (Section 4.4), in order to guide future observations, which parameters should be measured to better constrain the model. Comment also on model uncertainties in vertical profiles (e.g. T, RH, wind speed, sublimation rate) and drift layer height, and how they would impact on the main conclusions.

SPECIFIC COMMENTS

L142 Please provide detail on the model parameters for snow particles (size, shape) used in MAR.

L145 '... the drifting-snow scheme simulates erosion at every grid cell in which the modelled wind shear exceeds a threshold value depending on the local surface snow density.' What is this threshold value? How is it parameterised? Does snow particle size play a role? Please expand.

L231 'snow particle ratio'; do you mean here snow particle mixing ratio (mass of suspended snow particles to that of dry air)? Please clarify.

L243/Fig.3 Expand explanation - I assume the vertical maxima in SWnet (Fig.3d) reflect the diurnal cycle, and a small reduction is seen <100m on 3 Oct compared to 2 Oct, but the impact of drift snow on LWnet is only noticeable below 50m (Fig.3c). Why is that? And is it consistent with estimated drift snow layer heights during that time?

L278 'Drifting snow modifies the seasonal values of incoming radiative fluxes by enhancing LWD and decreasing SWD (Fig. 4 (e) and (g)). ' - the latter does not seems to be supported by Fig.4g, both model scenarios plot on top of each other, please clarify.

Fig.3b What is the averaging period simulated sublimation rate refers to? Per 30min or per hour? Are these values consistent with observations existing elsewhere in Antarctica?

TECHNICAL COMMENTS

Fig.3l Place legend outside the figure panel.

Fig.5a,c The grey shaded area to illustrate RMSE is missing.

REFERENCES

Déry, S. J. and Yau, M.: A Bulk Blowing Snow Model, Bound.-Lay. Meteorol., 93, 237–251, doi:10.1023/A:1002065615856,1999.

King, J. C. and Anderson, P. S. and Mann, G. W.: The seasonal cycle of sublimation at Halley, Antarctica, 47, 156, p1-8, J. Glaciol., 2001.

King, J. C. and Anderson, P. S. and Smith, M. C. and Mobbs, S. D.: The surface energy and mass balance at Halley, Antarctica during winter, 101, D14, J. Geophys. Res., 1996.

---

## Author Comment (AC1) · 14 Apr 2021

**Review: Sensitivity of the surface energy budget to drifting snow as simulated by MAR in coastal Adelie Land, Antarctica**

**General notes**

Please see our answer to R1.Q3 ("Reviewer 1, question 3") and R2.Q3. New model simulations have been computed, consequently all statistics presented in the original manuscript have undergone modifications. Please find at the end of the review the updated figures and tables. The text in the manuscript has been updated accordingly.

**Reviewer 1**

This paper examines the effects of snowdrift on Antarctica in a regional climate model with and without snowdrift physics included, and verified with local observations Adelie Land. Overall it is an interesting paper with useful results, showing that the inclusion of snowdrift physics considerably reduces the bias with observations. I recommend publication provided that the following issues are satisfactorily addressed.

We would like to thank Reviewer 1 for making a constructive review which we believe helped us improve the manuscript and more precisely the scientific discussion involving wind speed modifications.

**R1.Q1**
1. In places, the grammar is quite poor. For instance, there are many errors in the use of plural/singular. I strongly recommend to have the manuscript proofread and check by a native speaker or the like.

We apologize for the grammar errors. We carefully proofread the paper, made the necessary corrections when required, and improved the syntax. Please find our corrections in the track-change manuscript.

**R1.Q2**
2. In the model description section, I totally miss how snowdrift affects the momentum balance of the boundary layer. I suspect that the enhanced drag of saltating particles is included (as it should be), but is the buoyancy effect of the suspending particles also included? I recommend adding information on how the model handles these physics,since the authors do show results on wind speeds, which are interesting (see below).

Three processes drive modifications in the momentum balance when accounting for drifting snow in MAR.

Firstly, the weight of suspended snow particles (referred to as "loading" in Fig. S4 in the original manuscript), which increases the density of the air, is accounted for by including the contribution of snow particles in the computation of the potential virtual temperature (Gallée et al., 2001).

Secondly, as atmospheric sublimation contributes explicitly to the energy and mass budget of each model vertical level in which it occurs, additional sublimation of drifting-snow particles lowers the air temperature by latent heat consumption and contributes to an increase in air density and atmospheric stability. The increase in air density enhances the along-slope pressure gradient force, and is a positive feedback in katabatic flows (Kodama et al., 1985; Gallée and Pettré 1998). A sensitivity experiment is presented in Fig. R1, following the framework detailed in R1.Q3, and demonstrates that decreased potential temperatures in MAR-DR due to sublimation-induced cooling are responsible for an increase in katabatic forcing.

Finally, the drag of saltating particles is not explicitly taken into account in the current version of the model as it is still in development for further studies (see our answer to R1.Q3 regarding modification of statements made in the original manuscript regarding this assumption). However, with the new modifications included in the drifting-snow scheme (please also refer to R1.Q3 for further details), the modeled roughness length for momentum z0 has been calibrated to reproduce the observed seasonal variation in z0, derived from wind speed profiles at D17 (Amory et al., 2020).

We suggest to add the following sentences to the model description (Sect. 2.2 in the original manuscript), after modifications related to R2.Q1:

*"Ultimately, the momentum balance of the boundary layer is mainly affected through three distinct processes when accounting for drifting snow in MAR.*

*Firstly, the increase in air density due to the weight of suspended snow, which is accounted for in the model by including the contribution of suspended snow in the computation of the potential virtual temperature (Gallée et al., 2001), is inherently amplified when eroded particles contribute to the airborne snow mass.*

*Secondly, drifting-snow sublimation and subsequent cooling of the atmosphere is computed at each model vertical level and contributes to increase air density and atmospheric stability, which enhances the along-slope pressure gradient force and is a positive feedback in katabatic flows (Kodama et al., 1985; Gallée and Pettré 1998).*

*Finally, the aerodynamic roughness length z0 is computed following a relationship previously fitted on observed z0 values in order to take into account the seasonality of surface roughness in a drifting-snow climate as observed in Adelie Land (Amory et al., 2020). More precisely, z0 is computed as a function of air temperature (for temperature above -20°C) and set to a constant value of 2 10 ^-4 m representative of inland conditions (Vignon et al., 2017) for temperatures below -20°C."*

R1.Q3

3. The effects on wind speed. These are quite interesting and this should be considerably expanded. With snowdrift, wind speed reduces at the surface (due to drag of saltating particles), but increases higher up (owing to stronger cooling associated with enhanced snowdrift sublimation, and the consequent reinforced katabatic forcing). However, a detailed analysis, including a 2D framework, is lacking. Now this is mentioned casually in two sentences (lines 345-347), but this topic warrants an in-depth discussion. One possibility is to include high-level winds in a panel of Fig. 7, and add a discussion on the snowdrift-induced effects of surface drag and katabatic forcing as mentioned above. Another option is to look at the momentum budget of the boundary-layer, and evaluate the friction and buoyancy/katabatic forcing terms. This will shed more light into the (spatial, vertical) variations in e.g. wind speed changes caused by the inclusion of snowdrift.

Following your comments, a deeper and more precise analysis of the model and its core modules was led in order to identify which parameterization could modify directly or indirectly wind speed when drifting snow is activated (see our previous answer R1.Q2).

Firstly, we would like to note that the drifting-snow module should have taken into account a parameterization of the aerodynamic roughness length z0, which has been accidentally kept disabled. As mentioned above, this parameterization was initially developed in order to describe the seasonality of surface roughness in a drifting-snow climate as observed in Adelie Land. We refer to R1.Q2 for a detailed description of the parameterization. We consequently relaunched new simulations with the modified parameterization for z0 in the MAR-DR simulation (in addition to a correction of the computation of sublimation rates for both simulations, see our answer to R2.Q3) and the updated results are now reported.

The corrected z0 shows little influence on the presented results and does not alter the main conclusion despite modifications of the wind speed at the lowest vertical levels of the model. Indeed former results exhibited a small decrease in mean wind speed at the lowest model level (-0.3 m/s) between MAR-DR and MAR-nDR. Such a behavior is not retrieved in the corrected runs where wind speed increases at all model vertical levels in MAR-DR compared to MAR-nDR. As z0 depends on temperature, the air temperature

cooling due to enhanced atmospheric sublimation produces lower z0 values responsible for an increase in wind speed in the lower atmosphere compared to the simulation without drifting snow. Additionally cooling of the atmosphere with drifting-snow sublimation also increases wind speed through an increase in katabatic forcing.

As a consequence, we would like to correct a sentence line 346 in the original version to specify that the consumption of turbulent kinetic energy in order to maintain snow particles in suspension was actually not explicitly taken into account. However, we would like to point out that as the z0 parameterization was calibrated on observed values in Adelie Land, including observations collected during drifting snow events, the z0 computed in the corrected results partly take into account this effect.

We suggest to modify L345-347 to:

*"This effect could be moderated at the lowest vertical model levels by surface-atmosphere interactions, such as the surface drag responsible for a decrease in wind speed"*

Identifying corresponding parameterizations that could influence wind speed and quantifying their marginal effect is a relatively complex task as wind speed is a variable resulting from the interdependence of multiple other variables called at different locations in the model routines. We agree with the reviewer's that wind speed modifications and drifting-snow influence on momentum budget would benefit further investigations. We decide here to frame further discussion on the influence of drifting-snow sublimation on changes in the katabatic pressure gradient force only. Please find below additional information we suggest to add to the supplementary material (including Fig. R2).

*"Mahrt (1982) and van den Broeke and van Lipzig (2003) propose a framework to decompose the downslope momentum budget terms along a low inclination straight slope. This strategy is well suited for studying katabatic wind regimes (e.g. van Angelen et al 2011, van den Broeke et al. 2002, van den Broeke and van Lipzig 2003) and thus the influence of drifting snow on the katabatic forcing at D17. KAT (eq. R1) designates the downslope momentum budget term related to the katabatic pressure gradient force, $\theta_0$ is the background potential temperature, g is the standard acceleration due to gravity (9.81 m s-2) and α is the slope.*

$$KAT \ = \ \frac{g}{\theta_0}\Delta_\theta sin(\alpha) \qquad Eq. \ R1$$

*KAT results from a potential temperature deficit $\Delta_\theta$ between the air potential temperature and the background potential temperature, representative of a potential temperature out of the gravity flow (Marht 1982). The latter is obtained by the usual assumption of a linear behavior of potential temperature from the free atmosphere down to the lowest model vertical level. Mean vertical profile of $\Delta_\theta$ are reported on Fig. R1 at D17.*

*Drifting-snow sublimation cools the low atmosphere and increases temperature deficits in MAR-DR. This results in an increased katabatic forcing in the computation of the momentum budget in the model, and favors increasing wind speeds in the downslope direction (Eq. R1).*

[Figure]

*Figure R1. Sensibility of mean$\Delta_\theta$to the drifting-snow scheme computed with half-hourly outputs between 2010 and 2019. $\Delta_\theta$ is the potential temperature perturbation due to the presence of a gravity (katabatic) flow and is calculated accordingly to van den Broeke et van Lipzig (2003).$\Delta_\theta$decreases in MAR-DR due to drifting-snow sublimation and subsequent cooling of the atmosphere, and is responsible for the enhanced katabatic forcing in the lower boundary layer."*

We would like to summarize this discussion in the main material (L340 to 347) as follow:

*"Wind speed increases in MAR-DR compared to MAR-nDR at D17 (Fig. 6 (g), (h)). The largest increases are found at the sixth and seventh vertical levels (38m and 67m), near the level experiencing maximum sublimation (fourth vertical model level, 12 m). As already suggested (e.g., Kodama et al., 1985), wind speed can increase during drifting snow events because of increased density of the air-snow mixture and an increased stable thermal stratification (Fig. 6 (a)) caused by the atmospheric sublimation-induced cooling, which is a positive feedback on a sloping surface due to the gravitational nature of katabatic winds (Bintanja, 2000). This effect could be moderated at the lowest vertical model levels by surface-atmosphere interactions, such as the surface drag responsible for*

*a decrease of the wind speed Further analysis reveals that incorporating snow particles in the calculation of the virtual potential temperature, in order to simulate the contribution of snow particles to the air density has a small impact on wind speed in MAR-DR (Fig. S4). Finally, a supplementary analysis (Fig. R2) on the sensitivity of the katabatic forcing term to the inclusion of drifting snow is proposed through a computation of the potential temperature deficit in the low-atmosphere at D17, following van den Broeke et van Lipzig (2003). Decreasing temperatures with increasing drifting-snow sublimation modify mean potential temperature in the boundary layer. Such modifications are responsible for an increase in katabatic forcing in MAR-DR."*

Note that the spatial distribution of wind speed changes induced by drifting snow is presented in  Fig. S2 of the original manuscript. The differences that occured between MAR-DR and MAR-nDR are also retrieved at the scale of  the integration domain. Differences between MAR-DR and MAR-nDR occurring over the all integration domain are not prominent near the surface but peak higher up in the katabatic layer, in accordance with Fig. 6.

Finally, the modified wind values from the corrected runs are reported
both in the tables (Tables 1 and 2) presented in the review and in the manuscript text.

*R1.Q4*
4. The changed momentum budget and wind speed feed back on the magnitude of snowdrift transport, saltation and suspension. In essence, the stronger winds in the boundary layer in the case of snowdrifts will enhance snowdrifts (sublimation, suspension), etc. This feedback is worth exploring, as it may vary in sign depending on distance to the surface; this model provides the perfect means to do so.

We followed the reviewer advices and propose to add the following complementary analysis to the main material at the end of the paragraph line 347:

*"Moreover, higher wind speeds have the potential to (i) erode more snow, (ii) advect drifting-snow particles at higher elevations in a warmer and drier environment through turbulent mixing, (iii) increase the residence time of drifting-snow particles in the atmosphere. Consequently, higher wind speeds trigger three factors that could potentially favor more atmospheric sublimation and constitute a positive feedback. We explore this feedback in Fig. R2 (a) where MAR-DR drifting-snow fluxes are computed at each model vertical level and shown as monthly averages. Additionally, we performed the same computation by replacing the wind speed with the one from the simulation MAR-nDR, which is on average lower than in the MAR-DR simulation. We aim here at quantifying the absence of the positive feedback of sublimation on wind speed and its impact on drifting-snow fluxes. Fig. R2 shows that stronger  wind speeds reinforced by additional sublimation in the MAR-DR simulation are responsible for an increase in drifting-snow*

*fluxes. Such drifting-snow fluxes are correlated with atmospheric sublimation in a logarithmic fashion (Fig. R2 (b)): higher wind speeds induce higher drifting-snow fluxes which are in turn responsible for enhanced atmospheric sublimation. Enhancement of sublimation is however limited by the self-limiting feedback of sublimation (Bintanja, 2001), the latter being dependent on the under-saturation of the ambient environment (see colorbar on Fig. 3 (b)). Ultimately, our simulations suggest that the feedback of increased wind speed on atmospheric sublimation could be all the more important at higher elevations (ex: sixth model vertical level, 37m) where the atmospheric sublimation potential is more sensitive to increases in drifting-snow fluxes due to a lower relative humidity.*

[Figure]

*Figure R2. (a) Comparison between drifting-snow fluxes in the atmosphere calculated in MAR-DR using usual wind speed values (y axis), or using wind speed values retrieved from the MAR-nDR simulation (x axis). The latter is done to approximate drifting-snow fluxes without accounting for the impact of drifting-snow sublimation on wind speeds. All fluxes are monthly averaged values over the period 2010-2018, computed at each of the 10 lowest model vertical levels. The black line denotes the best linear regression. Taking into account the atmospheric sublimation feedback on wind speed enhances drifting-snow fluxes.*

*(b) Atmospheric sublimation as a function of drifting-snow fluxes for the 10 first model vertical levels. Values are averaged yearly to denote the model vertical level elevation (black text). Annual atmospheric sublimation rates are expressed in kg of sublimated snow mass per kg of moist air. The colorbar indicates yearly averaged relative humidity at the considered level. Enhanced drifting-snow fluxes are responsible for increased atmospheric sublimation, until a plateau is reached. This plateau coincides with the occurrence of near-saturated environments, where additional sublimation is limited by the negative feedback of sublimation."*

**R1.Q5**
5. The comparison with other models should be greatly expanded. In section 4.3 there's a very brief discussion on RACMO, but this is insufficient. Readers would like to know in

much more detail how your model results agree or differ from those obtained by RACMO, including the underlying physics. Otherwise this is "just" another model that studies the effect of snowdrift. Much can be learned by differences among models,especially about governing processes.

We fully agree with the reviewer that important information can be derived from a complete model intercomparison. Referring to other simulations and other models allows one to understand how independent approaches and scientific choices can lead to converging or diverging results. As RACMO was the only regional model to date, and to our knowledge, to provide continent/region scale simulations in Antarctica on timescale close to decades with and without activating drifting snow physics, we decided to refer to the published results and model description to underline specific differences between the two models. We agree that the reader would benefit a further in depth model intercomparison (including RACMO and also other recent drifting-snow models, e.g. Luo et al., 2021) giving insights about the effect of drifting snow on both the surface and the atmosphere. But such a work should ideally be only done by comparing simulations performed under comparable conditions (e.g. same region, comparable horizontal and vertical resolution, same boundary forcing) and we believe such a work is beyond the scope of our study and would benefit a fully dedicated publication. To clarify the position of our study, we propose to delete the paragraph concerning comparisons between RACMO and MAR which is incomplete as it stands. A new section *"4.3 Current limitations"* replaces sections 4.3 and 4.4 in the original manuscript but includes a concise discussion on some key features that differ between MAR and RACMO. The new section also includes modifications relative to points raised by Reviewer 2.

[revised manuscript text omitted]

King, J. C., Anderson, P. S., Smith, M. C., & Mobbs, S. D. (1996). The surface energy and mass balance at Halley, Antarctica during winter. Journal of Geophysical Research: Atmospheres, 101(D14), 19119-19128.

King, J. C., Anderson, P. S., & Mann, G. W. (2001). The seasonal cycle of sublimation at Halley, Antarctica. Journal of Glaciology, 47(156), 1-8.

Kodama, Y., Wendler, G., & Ishikawa, N. (1989). The diurnal variation of the boundary layer in summer in Adélie Land, eastern Antarctica. Journal of Applied Meteorology and Climatology, 28(1), 16-24.

Lenaerts, J. T. M., & Van den Broeke, M. R. (2012). Modeling drifting snow in Antarctica with a regional climate model: 2. Results. Journal of Geophysical Research: Atmospheres, 117(D5).

Lesins, G., Bourdages, L., Duck, T. J., Drummond, J. R., Eloranta, E. W., & Walden, V. P. (2009). Large surface radiative forcing from topographic blowing snow residuals measured in the High Arctic at Eureka. Atmospheric Chemistry and Physics, 9(6), 1847-1862.

Lin, Y. L., Farley, R. D., & Orville, H. D. (1983). Bulk parameterization of the snow field in a cloud model. Journal of Applied Meteorology and climatology, 22(6), 1065-1092.

Locatelli, J. D., & Hobbs, P. V. (1974). Fall speeds and masses of solid precipitation particles. Journal of Geophysical Research, 79(15), 2185-2197.

Luo, L., Zhang, J., Hock, R., & Yao, Y. (2021). Case Study of Blowing Snow Impacts on the Antarctic Peninsula Lower Atmosphere and Surface Simulated With a Snow/Ice Enhanced WRF Model. Journal of Geophysical Research: Atmospheres, 126(2), e2020JD033936.

Mahesh, A., Eager, R., Campbell, J. R., & Spinhirne, J. D. (2003). Observations of blowing snow at the South Pole. Journal of Geophysical Research: Atmospheres, 108(D22).

Mahrt, L. (1982). Momentum balance of gravity flows. Journal of Atmospheric Sciences, 39(12), 2701-2711.

Naaim-Bouvet, F., Guyomarc'H, G., Bellot, H., Durand, Y., Naaim, M., Vionnet, V., ... & Prokop, A. (2013, October). Lac Blanc Pass: a natural wind-tunnel for studying drifting snow at 2700ma. sl. In International Snow Science Workshop (ISSW) (pp. p-1332). Irstea, ANENA, Meteo France.

Palm, S. P., Kayetha, V., Yang, Y., & Pauly, R. (2017). Blowing snow sublimation and transport over Antarctica from 11 years of CALIPSO observations. The Cryosphere, 11(6), 2555-2569.

Uppala, S. M., Kållberg, P. W., Simmons, A. J., Andrae, U., Bechtold, V. D. C., Fiorino, M., ... & Woollen, J. (2005). The ERA‑40 re‑analysis. Quarterly Journal of the Royal Meteorological Society: A journal of the atmospheric sciences, applied meteorology and physical oceanography, 131(612), 2961-3012.

Vignon, E., Genthon, C., Barral, H., Amory, C., Picard, G., Gallée, H., ... & Argentini, S. (2017). Momentum-and heat-flux parametrization at Dome C, Antarctica: A sensitivity study. Boundary-Layer Meteorology, 162(2), 341-367.

van Angelen, J. H., Van den Broeke, M. R., & Van de Berg, W. J. (2011). Momentum budget of the atmospheric boundary layer over the Greenland ice sheet and its surrounding seas. Journal of Geophysical Research: Atmospheres, 116(D10).

van den Broeke, M. R., Van Lipzig, N. P. M., & Van Meijgaard, E. (2002). Momentum budget of the East Antarctic atmospheric boundary layer: Results of a regional climate model. Journal of the Atmospheric Sciences, 59(21), 3117-3129.

van den Broeke, M. R., & Van Lipzig, N. P. M. (2003). Factors controlling the near-surface wind field in Antarctica. Monthly Weather Review, 131(4), 733-743.

Yamanouchi, T., & Kawaguchi, S. (1984). Longwave radiation balance under a strong surface inversion in the katabatic wind zone, Antarctica. Journal of Geophysical Research: Atmospheres, 89(D7), 11771-11778.

Yang, Y., Palm, S. P., Marshak, A., Wu, D. L., Yu, H., & Fu, Q. (2014). First satellite‑detected perturbations of outgoing longwave radiation associated with blowing snow events over Antarctica. Geophysical Research Letters, 41(2), 730-735.

---

## Author Comment (AC2) · 14 Apr 2021

**Review: Sensitivity of the surface energy budget to drifting snow as simulated by MAR in coastal Adelie Land, Antarctica**

**General notes**

Please see our answer to R1.Q3 ("Reviewer 1, question 3") and R2.Q3. New model simulations have been computed, consequently all statistics presented in the original manuscript have undergone modifications. Please find at the end of the review the updated figures and tables. The text in the manuscript has been updated accordingly.

**Reviewer 2**

This is an interesting study on the impact of drifting and blowing snow on boundary layer meteorology, surface radiation and energy balance in Terre Adélie, a region on the slopes of the East Antarctic ice sheet. A 9-yr time series (2010-18) of observations at a site near the coast is used to validate the regional climate model MAR. Methodology, presentation and discussion of results as well as the conclusions are sound. One of the main findings is that sublimation of drifting-snow particles leads at the surface to a reduction in sensible and latent heat exchange, which is compensated by an increase in net radiative forcing. While the net impact on total surface energy budget, and therefore surface temperature, is minimal, structure of the lower atmosphere is modified, which needs to be resolved in climate models to understand impact of warming on air-surface interactions and boundary layer meteorology. I have only minor comments, and recommend publication after they have been addressed.

We thank reviewer 2 for making a positive review and instructive comments. Please find below our response to each of the points raised in the review.

**R2.Q1**
1. It would be useful to get more detail on how sublimation rates are computed in MAR and how reliable they are. What are the model assumptions (snow particle size distribution and shape).

The sublimation in MAR is distinguished between surface sublimation and sublimation of airborne particles (including both cloud-originating particles and drifting-snow particles raised from the surface), we suggest to modify and complete paragraph line 143 as follow:

*"[...]MAR is coupled to the surface scheme SISVAT (Soil Ice Snow Vegetation Atmosphere Transfer; De Ridder and Gallée (1998), Gallée and Duynkerke (1997), Gallée et al. (2001)), which handles energy and mass transfer between the atmosphere and the surface, and includes a multi layer snow/ice model representing snow properties (dendricity, sphericity and size) taken from an early version of the CROCUS snow model (Brun et al. 1992). Surface sublimation (which is distinguished in the model from atmospheric sublimation) and latent heat exchanges at the surface are computed following a bulk flux formulation in SISVAT.*

*MAR includes a drifting-snow scheme originally described in Gallée et al. (2001). A detailed description of MARv3.11 latest version (including updates, changes relative to the original version and interactions with the surface and the atmosphere) can be found in Amory et al. (2020). In brief, the drifting-snow scheme simulates erosion at every grid cell in which the modelled friction velocity exceeds a threshold value, u\*t, depending on local surface snow density. While former parameterisations of u\*t in the model did involve other snow microstructural properties such as snow grain shape and size (Gallée et al., 2001) for which observations are virtually non-existent in Antarctica, here the formulation for u\*t has been simplified and sensitivity parameters have been reduced to surface snow density only, a variable better observationally constrained (Amory et al. 2020). Once removed from the snowpack, eroded snow is mixed with the pre-existing windborne snow mass and advected to higher atmospheric levels and/or downwind grid cells by the turbulence and microphysical schemes. Interactions with the atmosphere are computed by the microphysical and the radiative transfer schemes. More particularly, atmospheric sublimation (including both cloud-originating particles and drifting-snow particles) is computed by the model microphysics (Gallée 1995). It incorporates a formulation for snow sublimation in the atmosphere (Lin et al. 1983). This formulation is based on the assumption of an exponential distribution for particle size and is a function of the air temperature, snow particles ratio and relative humidity (so that sublimation only occurs in a subsaturated environment, with respect to ice). It also considers snow particles as graupel-like snow of hexagonal type (Gallée et al. 1995, Locatelli and Hobbs 1974). Consequently, drifting-snow sublimation modifies the local humidity budget, the lower atmosphere stratification and moist air advection. Representing the contribution of drifting-snow layers to the atmospheric radiative forcing is accounted for in MAR by including suspended snow particles in the computation of cloud radiative properties (Gallée et Gorodetskaya, 2010). "*

This paragraph will be complemented by the additional information on how drifting snow affects the momentum budget in the boundary layer described in R1.Q2. Finally, we refer to our answer to R2.Q3 for a more detailed discussion on the reliability of sublimation rates.

1.2 Another model parameterisation of bulk sublimation rates from blowing snow by Déry and Yau (1999) uses the mean snow particle diameter. Is this the case in MAR and is particle diameter a sensitive parameter? If yes, future studies would gain by deploying next to an electro-acoustic sensor also an optical particle counter, to measure particle diameter as well as snow mass flux more accurately.

We refer to the previous question concerning the computation of atmospheric sublimation in MAR, which is not directly based on an assumption of a particle diameter but rather on an assumption of the distribution of particle size. However, we would like to inform the reviewer and our readers that an optical snow particle counter (SPC) has been deployed at D17 in January 2014 for a single drifting snow event to initiate first comparisons with FlowCapt sensors in Antarctic conditions, and assess the ability of the FlowCapt to measure drifting snow fluxes (S1, supplement of Amory 2020). Energy supply issues have so far limited the permanent use of an SPC at D17 . However, we do believe too that deploying an SPC at D17 in complement of acoustic measurement, such as already done at the Col du Lac Blanc in the french Alps (Naaim-Bouvet et al. 2013), would complete the already existing set of measurement devices and help future studies to assess modeling hypothesis and help the scientific community better understand drifting-snow physics. We refer to our answer to R2.Q3 including a discussion in the manuscript about the potential benefit of optical measurements at D17.

What are the uncertainties of calculated sublimation rates and how do calculations compared to existing observations in Antarctica (e.g. hourly blowing snow sublimation rates at Halley range 0.1-1 mm we/day (King et al., 1996))?

We thank the reviewer for that question that helped us spot a miscalculation in the initial model results, for which we apologize. The sublimation rates, as initially presented on Fig. 3 (b) and Fig. 6 (j), were not correctly computed. We corrected this issue, relaunched the simulations for the 9 year period (in addition to the correction related to R1.Q3), and we now report updated sublimation rates, computed following the method described in R2.Q1.

Sublimation rates are now expressed in [kg of sublimated snow / kg of moist air] instead of [mm w.e. year] on Fig. 3 and Fig. 6. Such a choice is justified to preserve the consistency between the model meteorological variables presented in Fig. 3. and Fig. 6 (wind speed, relative humidity, temperature…), which are representative of values averaged over each vertical model level. Such averaged values can not be directly compared to sublimation rates expressed in mm we (per unit of time), as such rates are representative of total sublimation, integrated over the thickness of each model vertical

level (a better comparison could be done by comparing e.g. sublimation rates in [mm w.e.] to latent heat release integrated over the thickness of the model vertical level). The unit now presented is appropriate for estimating a sublimation rate intensity, comparable between different models levels of distinct thicknesses to other variables as presented in Fig. 3 and Fig. 6.

Additionally, we estimated drifting-snow sublimation as the difference in atmospheric sublimation over the first 1000 m above ground between MAR-DR and MAR-nDR. The averaged drifting-snow sublimation over the integration domain is 606 mm we/year and is locally higher at D17 with 719 mm we/year. Such estimates are affected, among other, by a source of uncertainty in the model microphysics: as the atmosphere contains more humidity in MAR-DR due to enhanced atmospheric sublimation in comparison with MAR-nDR (and as eroded snow particles can also act as nuclei particles), MAR-DR might simulate more clouds and associated precipitations than MAR-nDR. Newly formed cloud and/or precipitations could potentially sublimate, which could induce an overestimation of drifting-snow sublimation rates. A distinction between sublimation of snowfalls and eroded snow particles could be theoretically done in the model, but is not currently implemented in MAR. Drifting-snow sublimation rates presented above thus can not be directly interpreted as a surface mass balance component as they do include the sublimation of cloud-originating particles that have not reach the surface yet. .

King et al. (1996) reported lower estimates of drifting-snow sublimation using a sublimation model fitted on observed data at Halley station (mean values being typically from 1 to 2 order of magnitude lower, but peak values of the same order of magnitude). Similarly, King et al (2001) and Bintanja et Reijmer. (2001) report sublimations rates up to 50 mm we/year and 70 mm we/year at Halley and near Svea station in Dronning Maud Land (sometimes even including surface sublimation). Finally, Bintanja (1998) report higher values in Adelie Land (typically around 150 mm we/year), characterized by a strong spatial variability.

MAR-DR simulates stronger drifting-snow sublimation rates. However, Palm et al. 2017 estimated, using remotely sensed data, that drifting-snow sublimation rates could be up to 250 mm we/year in Adelie Land. Finally, Lenaerts and van den Broeke 2012 estimated drifting-snow sublimation in Adelie Land to be up from 150 to more than 300 mm we/year. Both Palm et al. 2017 and Lenaerts and van den Broeke 2012 reported a very high spatial variability in drifting-snow sublimation.

Important disparities between models/instruments used to estimate drifting-snow sublimation (e.g. maximum/minimum height above ground until/from which sublimation in computed) and climatic differences between sites where measures have been held exist: e.g. D17 is both windier and warmer than Halley (King et al. 2001), favoring more drifting-snow sublimation. Consequently, quantifying the potential (and probable) extent

to which MAR-DR overestimate drifting-snow sublimation could hardly be done by just comparing with pre-existing estimates with different methods and from other locations, and we believe would lie beyond the scope of our study (an independent study using the same model is currently being held on the entire continent to discuss and quantify drifting-snow sublimation).

Accordingly to our answer to Reviewer 1 R1.Q5, we suggest to delete paragraph 4.3 and modify paragraph 4.4 into *"4.3 Current limitations"*. Consequently, we suggest to summarize this discussion as follows (this discussion is inserted in the main text as presented in R1.Q5):

*"Drifting-snow sublimation, defined here by the difference in atmospheric sublimation between MAR-DR and MAR-nDR over the first 1000m above ground, equals on average 606 mm we/year on the all Adelie Land domain, with higher values reported at D17 (716 mm we/year). These rates are larger than previous in-situ estimates of drifting-snow sublimation held in distinct parts of the continent (King et al. 1996, King et al. 2001) where the climate differs from the windy and (relatively warm) conditions of coastal Adelie Land. However, Palm et al 2017, through remotely sensed data, and Lenaerts and van den Broeke 2012, by using a regional climate model, report sublimation rates in Adelie Land which are more in agreement with our model estimates, though still twice to three times lower. Finally, the inclusion of newly formed clouds in MAR-DR as discussed in Sect. 2.4 can contribute to the probable overestimation of drifting-snow sublimation rates in MAR-DR. The sublimation rates, as simulated by MAR, have not been yet directly compared to in-situ measurements, although indirect comparisons have been made through the evaluation of near-surface air relative humidity and temperature. Accounting for drifting-snow sublimation in the present study has proven useful to modify the relative humidity of the lower atmosphere and help the model matching with observed relative humidity from a timescale of a single event (Fig. 2) to a seasonal scale (Fig. 3). The deployment of eddy-covariance systems including highly sensitive hygrometers could provide complementary atmospheric sublimation estimates to evaluate model simulations during calm to moderate conditions. However, using eddy-covariance devices during strong drifting-snow episodes remains a challenge as drifting-snow particles alter the observed signal and limit their use in Adelie Land (e.g. Bintanja, 2001). Moreover, including drifting snow in MAR shows large impacts on turbulent fluxes which compensate (and sometimes slightly override, e.g. at D17) modifications in radiative fluxes. Such a compensation also needs to be evaluated through comparison with direct in situ measurements of latent and sensible heat fluxes during drifting-snow occurrences to determine if MAR-DR simulates (more) realistic turbulent heat exchanges at the surface. Modeling hypothesis regarding drifting-snow particle distribution and subsequent sublimation rates could be better constrained using information derived from in-situ optical measurements (e.g. Naaim-Bouvet 2013)"*

2. The limitations of the current evaluation method needs to be expanded (Section 4.4), in order to guide future observations, which parameters should be measured to better constrain the model. Comment also on model uncertainties in vertical profiles (e.g. T, RH, wind speed, sublimation rate) and drift layer height, and how they would impact on the main conclusions.

We fully agree that the reader should be aware of the scope of validity of the results, its assumptions and the uncertainties associated. This discussion will be added to section 4.3 (see R1.Q5).

*"The vertical profiles presented in Fig. 6 have only been evaluated at 2 m. a.g.l. therefore, the behavior of the model, and the eventual benefit of accounting for drifting snow in order to capture more realistic atmospheric dynamics in the lower atmosphere still needs to be assessed. Daily radio soundings are operated at the closeby permanent station of Dumont D'Urville. However, sufficient climatic disparity exists between D17 location, situated on the marginal slope of the Antarctic continent and Dumont d'Urville station, situated beyond the continent boundaries on an island approximately 15 kilometers northeast of D17. Nevertheless a good agreement with observed values for several meteorological variables (wind speed, relative humidity, temperature, drifting-snow fluxes, incoming and outgoing radiative fluxes) and the fact that the model is well constrained at its boundaries by global reanalysis is an argument in favour of firstly studying model outputs at the first vertical level and then exploring its behavior at higher altitudes. Dropsondes observations near D17 location or operation of radiosoundings from the ground at D17 would help assess model performance and uncertainties at higher elevation in complement of near-surface observations."*

We refer to line 401 for the discussion about the necessity to evaluate turbulent fluxes using methods complementary to the bulk/profile methods. Such methods, such as e.g. the deployment of eddy covariance measuring devices, are still limited by the presence of hydrometeors in the atmosphere during drifting-snow episodes (e.g. Bintanja, 2001) and highlight the difficulty to measure drifting-snow sublimation.

We propose to better discuss the current limitation of our estimation of drifting-snow layer heights by adding the following sentences (inserted in Sect 4.3, see R1.Q5):

*"Additionally, we introduced a method, based on CALIPSO observations to estimate the height of a drifting-snow layer using model outputs. This method allows us to derive an objective criterion concerning snow concentration in the atmosphere to determine the presence (or not) of a drifting-snow layer and its height, during specific meteorological conditions. This method is limited by the fact that it has only been developed for 8 years*

*of CALIPSO observations collected near D17. Future work could focus on other locations in Antarctica to improve the determination of the snow concentration threshold by gathering more remotely sensed observations to be compared to model simulations. This could ultimately lead to an evaluation of modeled drifting-snow layer heights using CALIPSO observations on a specific test dataset. Ultimately, the use of a grounded lidar at D17 could provide complementary information concerning the vertical structure of drifting-snow layers."*

Finally, we refer to our answer to R2.Q3 to underline current limitations in assessing the ability of MAR to simulate realistic sublimation rates.

SPECIFIC COMMENTS

*R2.Q5*
L142 Please provide detail on the model parameters for snow particles (size, shape) used in MAR.L145 '...the drifting-snow scheme simulates erosion at every grid cell in which the modelled wind shear exceeds a threshold value depending on the local surface snow density.' What is this threshold value? How is it parameterised? Does snow particle size play a role? Please expand.

The parameterisation of the threshold friction velocity for initiation of drifting snow $u_{*t}$ (m/s) is fully described in Amory et al. (2020**)** and is based only on surface snow density.

$$u_{*t} = u_{*t0} \, exp(\frac{\rho_i}{\rho_0} - \frac{\rho_i}{\rho_s}) \qquad \text{Eq. R2}$$

$$u_{*t0} = \frac{log(2.868) - log(1 + 0.625)}{0.085} C_D^{0.5} \qquad \text{Eq. R3}$$

$$C_D = \frac{u_*^2}{U^2} \qquad \text{Eq. R4}$$

where $\rho_s$ the surface snow density, $\rho_i$ the density of ice, $\rho_0$ the density of fresh snow, $C_D$ the drag coefficient for momentum, U the wind speed in the lowest model vertical level and $u_*$ the friction velocity. Please find more information in Amory et al. (2020).

Please find below further elements we suggest to add to the model description line 145:

*"In brief, the drifting-snow scheme simulates erosion at every grid cell in which the modelled friction velocity exceeds a threshold value, u\*t, depending on local surface snow density. While former parameterisations of u\*t in the model did involve other snow microstructural properties such as snow grain shape and size (Gallée et al., 2001) for which observations are virtually non-existent in Antarctica, here the formulation for u\*t*

*has been simplified and sensitivity parameters have been reduced to surface snow density only, a variable better observationally constrained (Amory et al. 2020)."*

*R2.Q6*
L231 'snow particle ratio'; do you mean here snow particle mixing ratio (mass of suspended snow particles to that of dry air)? Please clarify.

Here the  snow particle ratio refers to the snow particle specific ratio: the mass of suspended snow particles to the total mass of air including dry air, humidity and the mass of all other hydrometeors.

We included this specification L172, to the first appearance in the text of the term snow particle ratio:
*"[...] the snow particle ratio (the specific ratio, which equals the mass of snow particles per kg of air, including dry air, humidity and the mass of all other hydrometeors) [...]*

*R2.Q7*
L243/Fig.3 Expand explanation - I assume the vertical maxima in SWnet (Fig.3d) reflect the diurnal cycle, and a small reduction is seen <100m on 3 Oct compared to 2 Oct,but the impact of drift snow on LWnet is only noticeable below 50m (Fig.3c). Why is that? And is it consistent with estimated drift snow layer heights during that time?

Your assumption is right, the diurnal cycle is indeed retrieved in the SWnet representation in Fig. 3d (intense yellow areas, corresponding to high SWD periods). We intended to point out simultaneous modifications in both modeled SWnet and LWnet during a drifting snow event occurring on the 3rd of October 2017 (Fig. 3 (e)). As pointed out by the reviewer, a first visual analysis on Fig. 3d indicates that modifications in SWnet with elevation start at approximately 100m while LWnet modifications are only visible below 50m. Small variations of LW are not retrieved in the current visualization but a finer data analysis point out increases smaller than 5 W.m² up to 100m.

Furthermore, we would like to underline the fact that drifting-snow layer heights were not initially retrieved using observations based on radiative modifications of longwave and shortwave fluxes. Indeed, CALIPSO observations consist of lidar measurements and the algorithm used in MAR deals with snow concentrations. We would also like to point out that drifting-snow concentration in the atmosphere decreases exponentially with height (Fig. 6 (i)). As a consequence, as snow residence in the atmosphere drives radiative modifications, more important radiative effects can be expected when approaching the surface. Consequently we believe it is consistent to simulate drifting-snow layer heights (representative of snow concentrations in the atmosphere) at a specific elevation (Fig. 3 (a)) and retrieve the more important radiative effects at a close but different elevation (e.g.

Fig 3 (c ) and (d)). Finally, Fig. 3 suggests that the radiative modification with altitude, as observed on Fig. 3, depends on the type of radiation (shortwave and longwave) in MAR.

*R2.Q8*
L278 'Drifting snow modifies the seasonal values of incoming radiative fluxes by enhancing LWD and decreasing SWD (Fig. 4 (e) and (g)). ' - the latter does not seems to be supported by Fig.4g, both model scenarios plot on top of each other, please clarify.

SWD modifications with drifting-snow are visible on single specific events, such as presented on Fig. 3d, but such modifications are not sufficient to be retrieved in seasonal means at D17, as presented on Fig. 4g.  We suggest to modify sentence L278 to:

*"Drifting snow enhances the seasonal values of LWD (Fig. 4 (e)), but even if significant modifications in SWD can occur during specific events such as presented in Fig. 3d, the impact on seasonal averages is low (Fig. 4 g)."*

The seasonality of SWD can partially explain such a difference between larger increases in LWD (see Sect 3.3), which are retrieved in seasonal means, and decreases in SWD, which are not visible in Fig. 4. SWD are weak or null a large part of the year in the high latitudes of Adelie Land meanwhile longwave emission of drifting-snow particles remains positive all year long. Furthermore wind speeds are stronger in winter at D17 (Fig. 4 (b)) favoring stronger and more frequent drifting snow during that part of the year, and thus greater radiative effects on LWD (see Fig. 4 (e)). However, Fig. 8 (Fig. 7 in the original manuscript) indicates that, on a yearly basis, drifting snow induces significant decreases in SWD in locations experiencing more intense drifting snow than D17.

*R2.Q9*
Fig.3b What is the averaging period simulated sublimation rate refers to? Per 30min or per hour? Are these values consistent with observations existing elsewhere in Antarctica?

Fig. 3 (b) (corrected, see our answer to R2.Q3) reports the quantity of snow sublimated during 30 min (the time step of model outputs), and is expressed in kg of sublimated snow per kg of air (see R2.Q3). This is now specified in Fig. 3. In the modified Fig. 6 (j), sublimation rates are expressed in kg of sublimated snow per kg of air per year. Please see our answer to R2.Q3 for further detail on the unit used for sublimation rates and comparison to observed value in Antarctica.

TECHNICAL COMMENTS

Fig.3l Place legend outside the figure panel.

The legend location has been modified.

Fig.5a,c The grey shaded area to illustrate RMSE is missing.

For better readability, we propose to keep the figure as it is and thus to delete the mention concerning missing RMSE.

**Updated figures and tables**

[revised manuscript text omitted]

King, J. C., Anderson, P. S., Smith, M. C., & Mobbs, S. D. (1996). The surface energy and mass balance at Halley, Antarctica during winter. Journal of Geophysical Research: Atmospheres, 101(D14), 19119-19128.

King, J. C., Anderson, P. S., & Mann, G. W. (2001). The seasonal cycle of sublimation at Halley, Antarctica. Journal of Glaciology, 47(156), 1-8.

Kodama, Y., Wendler, G., & Ishikawa, N. (1989). The diurnal variation of the boundary layer in summer in Adélie Land, eastern Antarctica. Journal of Applied Meteorology and Climatology, 28(1), 16-24.

Lenaerts, J. T. M., & Van den Broeke, M. R. (2012). Modeling drifting snow in Antarctica with a regional climate model: 2. Results. Journal of Geophysical Research: Atmospheres, 117(D5).

Lesins, G., Bourdages, L., Duck, T. J., Drummond, J. R., Eloranta, E. W., & Walden, V. P. (2009). Large surface radiative forcing from topographic blowing snow residuals measured in the High Arctic at Eureka. Atmospheric Chemistry and Physics, 9(6), 1847-1862.

Lin, Y. L., Farley, R. D., & Orville, H. D. (1983). Bulk parameterization of the snow field in a cloud model. Journal of Applied Meteorology and climatology, 22(6), 1065-1092.

Locatelli, J. D., & Hobbs, P. V. (1974). Fall speeds and masses of solid precipitation particles. Journal of Geophysical Research, 79(15), 2185-2197.

Luo, L., Zhang, J., Hock, R., & Yao, Y. (2021). Case Study of Blowing Snow Impacts on the Antarctic Peninsula Lower Atmosphere and Surface Simulated With a Snow/Ice Enhanced WRF Model. Journal of Geophysical Research: Atmospheres, 126(2), e2020JD033936.

Mahesh, A., Eager, R., Campbell, J. R., & Spinhirne, J. D. (2003). Observations of blowing snow at the South Pole. Journal of Geophysical Research: Atmospheres, 108(D22).

Mahrt, L. (1982). Momentum balance of gravity flows. Journal of Atmospheric Sciences, 39(12), 2701-2711.

Naaim-Bouvet, F., Guyomarc'H, G., Bellot, H., Durand, Y., Naaim, M., Vionnet, V., ... & Prokop, A. (2013, October). Lac Blanc Pass: a natural wind-tunnel for studying drifting snow at 2700ma. sl. In International Snow Science Workshop (ISSW) (pp. p-1332). Irstea, ANENA, Meteo France.

Palm, S. P., Kayetha, V., Yang, Y., & Pauly, R. (2017). Blowing snow sublimation and transport over Antarctica from 11 years of CALIPSO observations. The Cryosphere, 11(6), 2555-2569.

Uppala, S. M., Kållberg, P. W., Simmons, A. J., Andrae, U., Bechtold, V. D. C., Fiorino, M., ... & Woollen, J. (2005). The ERA‑40 re‑analysis. Quarterly Journal of the Royal Meteorological Society: A journal of the atmospheric sciences, applied meteorology and physical oceanography, 131(612), 2961-3012.

Vignon, E., Genthon, C., Barral, H., Amory, C., Picard, G., Gallée, H., ... & Argentini, S. (2017). Momentum-and heat-flux parametrization at Dome C, Antarctica: A sensitivity study. Boundary-Layer Meteorology, 162(2), 341-367.

van Angelen, J. H., Van den Broeke, M. R., & Van de Berg, W. J. (2011). Momentum budget of the atmospheric boundary layer over the Greenland ice sheet and its surrounding seas. Journal of Geophysical Research: Atmospheres, 116(D10).

van den Broeke, M. R., Van Lipzig, N. P. M., & Van Meijgaard, E. (2002). Momentum budget of the East Antarctic atmospheric boundary layer: Results of a regional climate model. Journal of the Atmospheric Sciences, 59(21), 3117-3129.

van den Broeke, M. R., & Van Lipzig, N. P. M. (2003). Factors controlling the near-surface wind field in Antarctica. Monthly Weather Review, 131(4), 733-743.

Yamanouchi, T., & Kawaguchi, S. (1984). Longwave radiation balance under a strong surface inversion in the katabatic wind zone, Antarctica. Journal of Geophysical Research: Atmospheres, 89(D7), 11771-11778.

Yang, Y., Palm, S. P., Marshak, A., Wu, D. L., Yu, H., & Fu, Q. (2014). First satellite‑detected perturbations of outgoing longwave radiation associated with blowing snow events over Antarctica. Geophysical Research Letters, 41(2), 730-735.

---

## Author Response (AR2)

Please find below an answer to additional comments reported in blue.

R2Q2 - In the literature two-parameter (shape and scale, product of which is mean particle diameter) gamma probability density functions were shown to give a reasonable fit to observed distributions of snow particle diameters (Budd, 1966; Schmidt, 1984), also used by Déry and Yau (1999). Apparently MAR is using a different parameterisation. For completeness and comparison, what is the distribution parameter lambda of the chosen exponential functions after Lin et al.(1983) used in this study, i.e. lambda or 1/lambda (=mean)? Presumably these have been fitted to observations in Antarctica, which ones? Please add this information on the chosen parameter, as it would help comparing to other models.

Please note that $\lambda_s$ (Eq. 3) doesn't constitute an input parameter of the model but is rather dynamically determined through computation of the snow particle ratio and the air density. However, the intercept parameter $n_0$ is empirically fixed to $3*10^8\ m^{-4}$. Note that this value has not been fitted to any observed data and is more generally relative to the horizontal resolution and size of integration domain adopted in this study. The average diameter of (drifting-) snow particles in MAR can indeed be estimated by computing $1/\lambda_s$ which typically yields values 2 to 6 times above the mean diameter reported in Déry and Yau (1999). Such a diameter also stands for any snow particle in the atmosphere since the model does not discriminate the source of snow particles (originating either from clouds or from the ground, Amory et al., 2021). Thus, we suggest modifying the main manuscript L154 as follows.

"This formulation is based on the assumption of an exponential distribution for particle size $n_s$ (Eq. 2):

$$n_s = n_0 exp(-\lambda_s D_s) \qquad Eq.\ 1$$

$n_0$ being a constant representing the intercept parameter of the distribution. $n_0$ is empirically determined and was set to $3*10^8\ m^{-4}$ in our study. $D_s$ corresponds to the particle diameter (expressed in m) and $\lambda_s$ the dispersion parameter (expressed in $m^{-1}$). $\lambda_s$ is determined as followed:

$$\lambda_s = \left(\frac{\pi*\rho*n_0}{\rho_a*q_s}\right)^{\frac{1}{4}} \qquad Eq\ 2.$$

with $\rho$ the snow particle density ($100\ kg\ m^{-3}$), $\rho_a$ is the air density ($kg\ m^{-3}$) and $q_s$ the snow particle ratio (expressed in kg of snow per kg of air).

Sublimation is then computed as a function of the air temperature, snow particle ratio and relative humidity, so that sublimation only occurs in a subsaturated environment, with respect to ice (Lin et al., 1983, their eq. 31, p. 1072)."

*References:*

Amory, C., Kittel, C., Le Toumelin, L., Agosta, C., Delhasse, A., Favier, V., & Fettweis, X. (2021). Performance of MAR (v3. 11) in simulating the drifting-snow climate and surface mass balance of Adélie Land, East Antarctica. *Geoscientific Model Development*, *14*(6), 3487-3510.

Déry, S. J., & Yau, M. K. (1999). A bulk blowing snow model. *Boundary-Layer Meteorology*, *93*(2), 237-251.

Lin, Y. L., Farley, R. D., & Orville, H. D. (1983). Bulk parameterization of the snow field in a cloud model. *Journal of Applied Meteorology and climatology*, *22*(6), 1065-1092.